# Asymmetric Prompt Weighting for Reinforcement Learning with Verifiable Rewards

Reinhard Heckel [* 1]   Mahdi Soltanolkotabi [* 2]   Christos Thrampoulidis [* 3]

## Abstract

Reinforcement learning with verifiable rewards has driven recent advances in LLM post-training, in particular for reasoning. Policy optimization algorithms generate a number of responses for a given prompt and then effectively weight the corresponding gradients depending on the rewards. The most popular algorithms including GRPO, DAPO, and RLOO focus on ambiguous prompts, i.e., prompts with intermediate success probability, while downgrading gradients with very easy and very hard prompts. In this paper, we consider asymmetric prompt weightings that assign higher weights to prompts with low, or even zero, empirical success probability. We find that asymmetric weighting particularly benefits from-scratch RL (as in R1-Zero), where training traverses a wide accuracy range, and less so in post-SFT RL where the model already starts at high accuracy. We also provide theory that characterizes prompt weights which minimize the time needed to raise success probability from an initial level to a target accuracy under a fixed update budget. In low-success regimes, where informative responses are rare and response cost dominates, these optimal weights become asymmetric, upweighting low success probabilities and thereby accelerating effective-time convergence.

## 1. Introduction

Recent advances for post-training LLMs have substantially improved their reasoning, math, and coding abilities. In particular, reinforcement learning with verifiable rewards (RLVR) that leverages simple–often binary–feedback, such as the correctness of an answer, have driven the recent progress in reasoning performance (Guo et al., 2025; Team et al., 2025a; Jaech et al., 2024).

Policy optimization algorithms for LLM post-training typically first sample a batch of prompts (e.g., math problems), second generate a set of responses for each prompt (e.g., solutions to math problems), third compute a reward (e.g., 0/1 depending if the solution is correct), and finally update the model based on the normalized rewards. The most popular algorithms including GRPO (Shao et al., 2024), DAPO (Yu et al., 2026), RLOO (Kool et al., 2020; Ahmadian et al., 2024) and variants thereof focus on ambiguous prompts, i.e., prompts with intermediate success probability.

In this paper, we propose asymmetric prompt weightings assigning higher weights to prompts with low success probability. Those weightings also assigns non-zero weight even when all completions receive zero reward, whereas GRPO and variants assign zero weight to such gradients, and thus fail to leverage these as training signals.

We perform two sets of experiments: First, we consider two from-scratch RL, R1-Zero-like (Guo et al., 2025) setups where we start from a model initially not performing well on reasoning, and we reach high performance through RL. For the first of these, we train the base model Qwen2.5-3B on a countdown task, and for the second we train the base model Llama-3.1 on GSM8K. For both setups, the base model starts below 0.02 accuracy and improves to about 0.8, and our asymmetric prompt weighting outperforms symmetric schemes such as GRPO, DAPO, and RLOO. Therefore, asymmetric weighting consistently improves performance in the from-scratch regime.

Second, we consider two post-SFT RL setups where the model is already good at reasoning through SFT, and RL is used to further improve performance. Specifically, we train Llama-3.2-3B-instruct on MATH (Hendrycks et al., 2021), and DeepSeek-R1- Distill-Qwen-1.5B on DAPO-math (Yu et al., 2026). In these settings, training begins from an intermediate success rate (0.3 and 0.4) and significantly improves (to 0.5 and 0.55), but we observe no meaningful difference between asymmetric and symmetric advantage weightings. In those post-SFT regime, asymmetric weighting yields no

---
[*]Equal contribution [1]Dept. of Computer Engineering, Technical University of Munich [2]Dept. of Electrical and Computer Engineering, University of Southern California [3]Dept. of Electrical and Computer Engineering, University of British Columbia. Correspondence to: Reinhard Heckel <reinhard.heckel@tum.de>, Mahdi Soltanolkotabi <soltanol@usc.edu>, Christos Thrampoulidis <cthrampo@ece.ubc.ca>.

*Proceedings of the 43rd International Conference on Machine Learning*, Seoul, South Korea. PMLR 306, 2026. Copyright 2026 by the author(s).

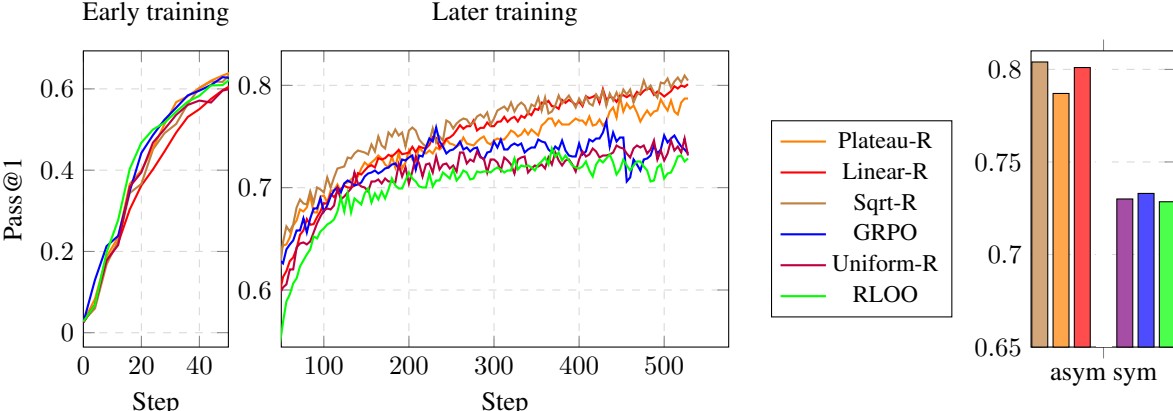

*Figure 1.* From-scratch RL: TinyZero. Test accuracy (i.e., fraction of correctly solved problems, Pass@1) during reinforcement learning. Starting from a score of 0.025 the algorithms based on asymmetric weighting (Linear-R, Plateau-R, and Sqrt-R) climb to about 0.8 while the symmetric weightings (RLOO, Uniform-R, and GRPO) only reach around 0.74.

additional gain but do not hurt performance.

Finally, we provide theory that characterizes prompt weights which minimize the time needed to raise success probability from an initial level to a target accuracy under a fixed update budget. In low-success regimes, where informative rollouts are rare and rollout cost dominates, these optimal weights become asymmetric, upweighting low success probabilities and thereby accelerating effective-time convergence.

### 1.1. Related work

There are a number of works that counteract GRPO's weak signal on hard prompts, or better leverage gradients from zero-reward rollouts.

Team et al. (2025b) used a prioritized sampling strategy where samples are reweighted by $1 - \hat{\rho}$, which effectively makes harder examples more likely to be sampled. Thrampoulidis et al. (2026) interprets this through surrogate reward maximization (cf. Section 2.1.2), thus connecting it to GRPO-modifications targeting Pass@K maximization (Chen et al., 2025; Mahdavi et al., 2025). While these forms of reweighting hard-examples have an effect analogous to an asymmetric prompt weighting, prompts corresponding to $\hat{\rho} = 0$ are still set to zero. In contrast, our work directly modifies the advantage estimators to ensure that prompts in the zero-success regime provide a boosted gradient signal.

Le et al. (2026) addresses zero-variance prompts by entropy-guided advantage shaping at the completion/token level, whereas we introduce prompt-level reweighting based on the success rate. Feng et al. (2025) reshapes learning based on how confident the model was at each mistake, whereas we reshape learning based on how hard the prompt currently is. Hong et al. (2025)'s 'asymmetric policy optimization' for multi-modal LLMs splits responses into positive and negative groups and applies different KL-based regularization schemes, while we adjust advantages. CoDaPO (Zhou et al., 2025) shares our motivation of counteracting GRPO's weak signal on hard prompts, but its difficulty reweighting is only indirectly tied to average performance and is applied jointly with an explicit confidence term, whereas our Linear-R uses a simple prompt-only weight. Wang et al. (2025) proposed an asymmetric token-level reweighting via policy ratios and advantage sign, whereas we study prompt-level reweighting as a function of success probability.

A recent result (Davis & Recht, 2025) shows that different weighting schemes can be interpreted as optimizing different surrogate objectives. Concurrently, Thrampoulidis et al. (2026) specifies an explicit recipe for deriving policy-gradient algorithms as weightings of RLOO starting from a surrogate reward. However, both perspectives leave open the practical question of which surrogate (equivalently, which weighting) should be preferred in different regimes. We address this directly: from a practitioner's viewpoint, we clearly demonstrate settings in which different weightings beat the dominant GRPO baseline; from a theory viewpoint, we use a policy-dynamics perspective to provide guidance on selecting weights that account for the prompt-difficulty distribution, showing in particular that asymmetric weights are most beneficial in the rare-success regime. Finally, while our Linear-R weighting is partly inspired by the log surrogate advocated by Davis & Recht (2025), their proposed implementation via rejection sampling differs from ours and is empirically less competitive (App. B.4 for a comparison).

## 2. Problem setup and background

Given pairs $(x, a)$ of problem $x$ (in form of a prompt) and (typically) a reference solution/answer $a$, our goal is to maximize the expected reward,

$$\mathbb{E}_{(x,a)} \left[ \mathbb{E}_{y \sim \pi_\theta(\cdot|x)} \left[ r(x, y, a) \right] \right], \tag{1}$$

where we take expectation over problems and associated reference solutions, $\pi_\theta$ is the policy (an LLM) parameterized by $\theta$ that we optimize over, $y$ is a response of the LLM to the prompt $x$, and $r$ is a reward for the response that is computed based on the reference answer $a$.

For example for mathematical reasoning, the reward checks whether the suggested solution in the completion $y$ is consistent with the reference solution $a$, and for code generation, the reward checks whether a suite of tests $a$ passes.

The most commonly used policy optimization algorithms for LLMs for RL with verifiable rewards follow the recipe:

1. Sample a batch of prompts $x_1, \ldots, x_B$ from a dataset.

2. For each prompt $x_j$, the current model $\theta_{\text{old}}$ generates $M$ responses (also called completions, rollouts, trajectories).

3. Compute a binary reward for each of the responses, for example with a verifier that tests whether a problem specified by $x_j$ has been solved correctly or not.

4. Update the model based on normalized rewards, called advantages, with one gradient step (on-policy training) or several gradient steps (off-policy training).

Policy optimization algorithms such as PPO (Schulman et al., 2017), REINFORCE, RLOO, and GRPO differ primarily in:

- **Advantage computation:** How rewards are normalized.

- **Loss aggregation:** Sample-average (all rollouts equal, e.g., GRPO), prompt-average (each prompt equal, e.g., DAPO), or token-average.

- **Off-policy handling:** On-policy training uses each batch for one update; reusing rollouts for multiple updates becomes off-policy. Algorithms address this mismatch via importance weighting/clipping/KL-penalties, or ignore it.

In this paper we focus on an on-policy setup, where each update is a single policy-gradient step based on per-prompt samples and their advantages.

As discussed below, several widely used policy-optimization methods for LLMs can be interpreted as inducing prompt-level weightings. For binary rewards, methods such as GRPO and RLOO yield weights that only depend on the average reward $\hat{\rho}_x$ of the responses for a prompt $x$ and are symmetric in that prompts that are easy (low $\hat{\rho}_x$) or hard (high $\hat{\rho}_x$) under the current policy receive smaller weight than prompts with intermediate success probability.

Motivated by the observation that learning can be bottle-necked by hard prompts, we propose and study **asymmetric**

prompt weights that deliberately assign higher weight to poorly-performing prompts (small $\hat{\rho}_x$). We find asymmetric reweighting improves learning in from-scratch RL setups.

## 2.1. Policy optimization for verifiable rewards

We consider rewards bounded in $[0, 1]$ for convenience. For a prompt $x$, let $y \sim \pi_\theta(\cdot|x)$ be the response and define the expected reward

$$\rho_x(\theta) = \mathbb{E}_{y \sim \pi_\theta(\cdot|x)}\left[r(y)\right], \tag{2}$$

where, for notational simplicity, we omit the dependence of the reward $r$ on a reference answer $a$ and the prompt $x$. The canonical per-prompt policy gradient direction is

$$\nabla_\theta \rho_x(\theta) = \mathbb{E}_{y \sim \pi_\theta(\cdot|x)}\left[r(y)\nabla_\theta \log \pi_\theta(y|x)\right].$$

We view policy-gradient updates as aggregating these per-prompt directions with prompt- and policy-dependent scalar weights $w_x(\theta)$, i.e.,

$$\nabla_\theta J(\theta) = \mathbb{E}_x\left[w_x(\theta)\nabla_\theta \rho_x(\theta)\right].$$

Different policy-optimization algorithms correspond to different choices of weights, and different finite-sample estimators $\hat{d}_x(\theta)$ of the per-prompt direction $\nabla_\theta \rho_x$. For example:

- Williams' REINFORCE (Williams, 1992) uses weight $w_x = 1$ and estimates the per-prompt direction $\nabla_\theta \rho_x$ as

$$\hat{d}_x(\theta) = \frac{1}{M}\sum_{i=1}^{M} r(y_i)\nabla_\theta \log \pi_\theta(y_i|x).$$

- REINFORCE leave-one-out (RLOO) (Kool et al., 2020; Ahmadian et al., 2024) also uses weight $w_x = 1$ but uses a leave-one-out estimate of the per-prompt direction:

$$\hat{d}_x(\theta) = \frac{1}{M}\sum_{i=1}^{M}\left(r(y_i) - \bar{r}_{-i}\right)\nabla_\theta \log \pi_\theta(y_i|x),$$

where $\bar{r}_{-i} = \frac{1}{M-1}\sum_{j \neq i} r(y_j)$ is the average reward of all completions but completion $y_i$.

- GRPO (Shao et al., 2024) uses weight $w_x = 1/\sigma_x$, where $\sigma_x^2$ is the empirical variance of the rewards $r(y_1), \ldots, r(y_M)$, and estimates the direction as

$$\hat{d}_x(\theta) = \frac{1}{M}\sum_{i=1}^{M}\left(r(y_i) - \hat{\rho}\right)\nabla_\theta \log \pi_\theta(y_i|x),$$

where $\hat{\rho}$ is the empirical average of rewards. Dr. GRPO (Liu et al., 2025) uses the same directional estimate, but weight $w_x = 1$.

### 2.1.1. BINARY REWARDS

For binary rewards, the direction estimates simplify. Define $\widehat{\nabla}_0 = \frac{1}{M_0} \sum_{i:r_i=0} \nabla_\theta \log \pi_\theta(y_i|x)$ and $\widehat{\nabla}_1 = \frac{1}{M_1} \sum_{i:r_i=1} \nabla_\theta \log \pi_\theta(y_i|x)$ as the average of the gradients with reward 0 and reward 1. Here, $M_0$ and $M_1$ are the number of rewards that are 0 and 1, and $r_i := r(y_i)$. With this notation, the gradient updates are:

- REINFORCE: $\widehat{G}_x(\theta) = w_x \cdot \hat{d}_x(\theta) = 1 \cdot \widehat{\rho}\, \widehat{\nabla}_1$.

- RLOO:

$$\widehat{G}_x(\theta) = w_x \cdot \hat{d}_x(\theta) = 1 \cdot \frac{M}{M-1} \hat{\rho}(1-\hat{\rho}) \left( \widehat{\nabla}_1 - \widehat{\nabla}_0 \right).$$

- GRPO:

$$\widehat{G}_x(\theta) = w_x \cdot \hat{d}_x(\theta) = \frac{1}{\sqrt{\hat{\rho}(1-\hat{\rho})}} \cdot \hat{\rho}(1-\hat{\rho}) \left( \widehat{\nabla}_1 - \widehat{\nabla}_0 \right).$$

For binary RLOO and GRPO, we can view $\hat{\rho}(1-\hat{\rho})$ and $\sqrt{\hat{\rho}(1-\hat{\rho})}$ as the effective prompt weight. Moreover, the GRPO direction is (up to the constant $\frac{M}{M-1}$) the same as for RLOO, only the weight $w_x$ is different.

For both RLOO and GRPO, prompts that are hard given the current policy ($\hat{\rho}$ close to zero) as well as prompts that are easy ($\hat{\rho}$ close to one) get de-emphasized. This makes sense, given the viewpoint that for hard prompts gradients are unreliable and for easy ones, we see diminishing returns.

### 2.1.2. SURROGATE REWARD PERSPECTIVE

Davis & Recht (2025); Thrampoulidis et al. (2026) noted that the different weights induce different surrogate-loss functions. In particular, REINFORCE and RLOO correspond to optimizing the expected reward $J(\theta) = \mathbb{E}_x[\rho_x]$, while GRPO corresponds to a surrogate objective of the form $J(\theta) = \mathbb{E}_x[F(\rho_x)]$, for $F(\rho_x) = 2\arcsin(\sqrt{\rho_x})$. This holds only for binary rewards; for non-binary rewards, no surrogate loss exists. Thrampoulidis et al. (2026) suggests designing binary-reward RLVR updates by choosing differentiable surrogate reward $F$ and up-weighting the RLOO gradient $\omega_x(\rho) \leftarrow F'(\rho)$. Here, we work directly with the weight function $\omega_x(\rho)$, which provides greater flexibility beyond the binary reward setting. More importantly, we propose concrete new weightings and provide detailed empirical and theoretical analysis showing precisely when and why asymmetric weights outperform existing baselines such as GRPO.

## 3. Policy optimization with asymmetric prompt weightings for binary rewards

As discussed, REINFORCE, RLOO, and GRPO leverage prompt weightings that can be viewed as focusing on ambiguous prompts, i.e., they assign high weights to prompt with intermediate success probability, while downgrading easy and hard ones, which can stabilize training, but may also lead to stagnation for hard prompts.

Here we propose a weighting that focuses on progress by upweighting prompts with low success probabilities, i.e., low values of $\rho_x$ under the current policy.

As introduced in the previous section, we focus on general RLVR algorithms with gradient updates for a given prompt $x$ consisting of a per-prompt weight $\omega_x$ and finite-sample estimate $\hat{d}_x(\theta)$ for the per-prompt direction $\nabla_\theta \rho_x$, i.e.,

$$\widehat{G}_x = \omega_x(\widehat{\rho}) \cdot \hat{d}_x(\theta). \tag{3}$$

We focus on the binary-reward case and take the finite-sample estimate $\hat{d}_x(\theta)$ for the per-prompt direction as the gradient direction of RLOO and GRPO:

$$\hat{d}_x(\theta) = \widehat{\rho}(1-\widehat{\rho}) \cdot \left( \widehat{\nabla}_1 - \widehat{\nabla}_0 \right). \tag{4}$$

Equivalently, written in the conventional advantages-form:

$$\widehat{G}_x = \frac{1}{M} \sum_{i=1}^{M} A_i \cdot \nabla_\theta \log \pi_\theta(y_i|x),$$
$$\text{with } A_i = \omega_x(\widehat{\rho}) \cdot (r_i - \widehat{\rho}). \tag{5}$$

To understand the benefit of asymmetric prompt weightings, we consider four weightings that all upweight prompts with low success probabilities relative to GRPO and RLOO, and with a naming that reflects the *effective weights* that they assign gradients (see Figure 2):

- **Linear-REINFORCE (Linear-R)**: We consider weighting $w_x(\rho) = \frac{1}{\rho}$, which focuses on failing prompts, and corresponds to the effective (linear) weight $1 - \rho$. In terms of advantages, in view of Equation (5), this corresponds to $A_i = \frac{1}{\rho}(r_i - \widehat{\rho})$. For wrong responses ($r_i = 0$) Linear-R assigns advantage $A_i = -1$; thus, the algorithm's overall gradient is non-zero even if $\widehat{\rho} = 0$.

- **Sqrt-REINFORCE (Sqrt-R)**: Consider the weight $w_x(\rho) = \frac{1}{\rho\sqrt{1-\rho}}$, which has the $1/\rho$ weighting dominating for small $\rho$, just like Linear-REINFORCE, and for $\rho$ close to one, behaves like the GRPO weighting. The effective weight is $\sqrt{1-\rho}$. Just like Linear-$R$, this assigns non-zero effective weight even if $\widehat{\rho} = 0$

- **Plateau-REINFORCE (Plateau-R):** Consider the weight

$$w_x(\rho) = \begin{cases} \frac{1}{2\rho(1-\rho)} & \rho < 1/2 \\ \frac{1}{\sqrt{\rho(1-\rho)}} & \rho \geq 1/2 \end{cases}. \tag{6}$$

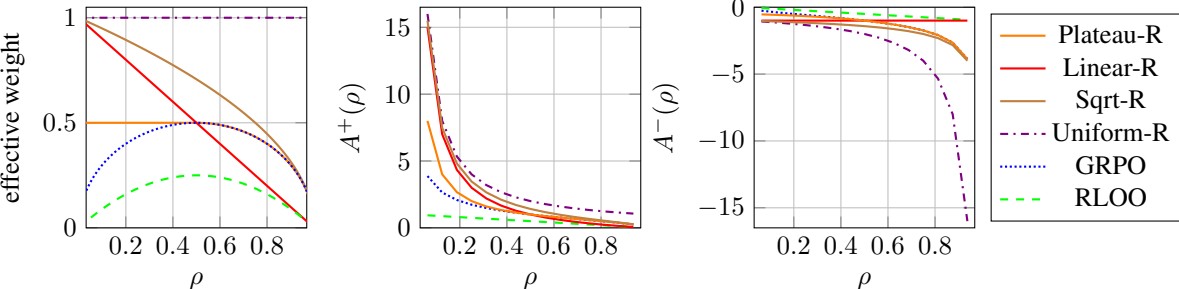

*Figure 2.* Effective weights $\omega_x(\rho) \cdot \rho(1-\rho)$ assigned to gradients, and advantages assigned to correct/wrong responses for the five considered prompt weightings. The range $1/32, \ldots, 31/32$ is shown since the number of rollouts $M$ is typically at most 32.

This gives an effective weight that is equal to that of GRPO for $\rho > 1/2$ and constant for smaller $\rho$. The weight function above corresponds to advantages $A_i = \omega_x(\rho) \cdot (r_i - \rho)$ which assign non-zero gradient weights $-1/2$ for prompts with $\rho = 0$.

- **Uniform-REINFORCE (Uniform-R):** Consider the weight $w_x(\rho) = \frac{1}{\rho(1-\rho)}$, which leads to an effective weight of $1$. We dub this Uniform-REINFORCE because it has effective weight $1$, since it cancels the factor $\rho(1-\rho)$, and thus weighs all prompts equally.

The weights assigned to gradients and corresponding advantages (for binary rewards) are summarized in Table 1 and illustrated in Figure 2.

In isolation, $\omega_x(\rho)$ can be undefined for $\rho = 0$ or $1$; however it multiplies with the direction and the *effective weight* $\omega_x(\rho) \cdot \rho \cdot (1-\rho)$ is always well-defined, including at the edge cases $\rho = 0, 1$; see Figure 2.

We also note that the benefit of asymmetric weighting does *not* reduce to simply taking larger gradient steps. While the proposed methods assign larger absolute advantages than GRPO at low $\hat{\rho}$, what matters is the *relative* weighting across prompts of different difficulty. For instance, as seen in Figure 2, at $\hat{\rho} = 1/8$ versus $\hat{\rho} = 7/8$, the ratio of positive advantages $A^+(\hat{\rho})$ is 49:1 for Linear-R but only 7:1 for GRPO. Scaling GRPO's learning rate by any constant preserves this ratio, leaving its relative emphasis across difficulty levels unchanged. In contrast, asymmetric weighting reshapes which prompts receive more gradient signal, a qualitatively different mechanism.

## 4. Experiments

We perform experiments on four different datasets and setups, each covering different regimes.

We first consider two from-scratch RL reasoning setups. Both setups are in a R1-Zero-like (Guo et al., 2025) regime where we start from a base model that initially does not perform well on reasoning, and eventually reach high performance through RL. The first setup is TinyZero (Pan et al., 2025) where the task is to solve a countdown and multiplica-

tion task. The second setup is GSM8K (Cobbe et al., 2021), a math reasoning task, where we start from a base model not trained or finetuned specifically for math. In both setups, the base model's average reward starts below 0.02 and advances to around 0.8. We find that our proposed asymmetric prompt weightings (Sqrt-R, Plateau-R, and Linear-R) outperform the non-symmetric ones (GRPO, RLOO, Uniform-R) in both cases.

Second, we consider two post-SFT RL setups where the model is already good at reasoning through SFT, and the goal of RL is to improve performance further. The first uses the DAPO-math dataset (Yu et al., 2026), and the second the MATH dataset (Hendrycks et al., 2021). In the two setups the average reward starts at 0.3 and 0.4 and reach 0.5 and 0.55, respectively. Unlike the from-scratch setting, we observe no notable difference among the advantage estimators GRPO, RLOO, Plateau-R, and Linear-R.

We use the SkyRL codebase (Cao et al., 2025), and have implemented our advantage estimators within that framework; experimental details are in the text and Appendix C.

### 4.1. From-scratch RL: TinyZero

We begin with the TinyZero setup proposed by Pan et al. (2025), training the Qwen2.5-3B base model to solve countdown and multiplication tasks. An illustrative prompt is:

```
<|im_start|>system
You are a helpful assistant. You first think
about the reasoning process in your mind
and then provides the user with the answer.
<|im_end|>
<|im_start|>user
Using the numbers [3,6,25,50,75,100], create
an equation that equals 952. You can use
basic arithmetic operations (+, -, *, /)
and each number can only be used once. Show
your work in <think> </think> tags. And
return the final answer in <answer>
</answer> tags, for example
<answer> (1 + 2) / 3 </answer>.
<|im_end|> <|im_start|>assistant
Let me solve this step by step.
<think>
```

We use a binary outcome reward only. Original

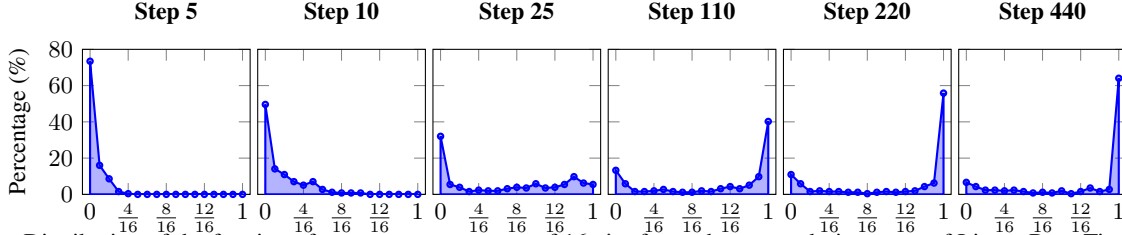

*Figure 3.* Distribution of the fraction of correct responses out of 16, $\hat{\rho}_x$, for each prompt during a run of Linear-R on TinyZero. At early steps, the distribution is heavily concentrated at low $\hat{\rho}$, reflecting that most prompts are difficult for the base model. Importantly, a substantial number of difficult prompts remain even at later steps (e.g., step 220). Linear-R continues to provide strong gradient signal for these prompts due to its weighting, unlike GRPO.

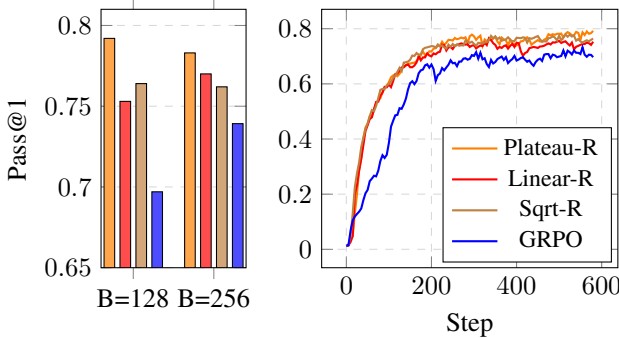

*Figure 4.* From-scratch RL: GSM8K. Test accuracy (i.e., fraction of correctly solved problems, Pass@1) during RL on the GSM8K benchmark for the Llama-3.1-8B (base) model. Asymmetric weighting (Plateau-R, Linear-R, Sqrt-R) outperform GRPO, demonstrating the benefit of up-weighting difficult prompts.

TinyZero (Pan et al., 2025) uses a format reward of 0.1 when the format is correct, in addition to an outcome reward; however, we find this unnecessary to reach good performance. We choose this setup because, as noted by (Pan et al., 2025), it is a minimal setup where one can experience the "Aha moment" reported in the R1-paper (Guo et al., 2025), where the model starts to have a productive thinking process. We choose $M = 16$ rollouts per prompt and a batch size of 512 (i.e., 512 distinct prompts, and for each prompt $M = 16$ completions).

Figure 1 shows the test accuracy (Pass@1) for different weightings: The model's initial performance is very low (about 0.025) and climbs to 0.8 for Linear-R and Plateau-R, compared to 0.74 for the other algorithms RLOO, GRPO, and Uniform-R. The asymmetric prompt weightings Plateau-R and Linear-R perform significantly better (6% higher Pass@1 score), and this improvement is reproducible: the plot shows three independent runs for Linear-R.

We hypothesize the following explanation for this behavior: All algorithms perform similar up to steps ∼150-200, where the score is ∼ 0.74. From here on, the asymmetric prompt weightings (Linear-R, Plateau-R, Sqrt-R) continue to improve to 0.8 while the symmetric ones (RLOO, GRPO, and Uniform-R) saturate. Figure 3 shows the distribution of the values of $\hat{\rho}$ (for Linear-R, though the other distributions are

essentially the same at Step 220): at Step 220 there are still examples with very low values for $\hat{\rho}$. GRPO and RLOO assign very small weights to the corresponding gradients (see Figure 2), as opposed to Linear-R, Plateau-R, Sqrt-R which assign higher weights and thus continue making progress on those examples, translating in overall better score.

Thus, upweighting gradients for low-$\rho$ prompts can translate in performance gains in settings where many prompts remain difficult to solve and thus have low $\rho$. In Section 5, we provide theoretical support for this finding by showing, under a tractable model of learning dynamics, that the Linear-R weighting is optimal in the low-$\rho$ regime.

Finally, recall that our asymmetric weightings (Linear-R, Plateau-R, Sqrt-R) assign non-zero advantages even if all completions have zero reward ($\hat{\rho} = 0$), unlike GRPO and RLOO. We conducted an ablation in which we set the advantage to zero whenever all completions failed; this hurt performance, see Appendix B.1.

### 4.2. From-scratch RL: GSM8K

Next, we consider the GSM8K dataset. We use the Llama-3.1-8B-base model (as opposed to Qwen base models as common in the literature), since Qwen models have been trained on data to improve their math performance, and we wish to understand what is learned through policy optimization algorithms with different weightings/advantages. We performed experiments with $M = 16$ completions, learning rate of 1e-6, and batch sizes of 128 and 256. The base model performs, as expected, relatively poorly initially, only achieving a Pass@1 rate of 1.3%. In this regime, most prompts have no correct completions ($\rho = 0$), and dominate the gradient for Plateau-R and Linear-R. We therefore set the advantages of Plateau-R and Linear-R for prompts with success rate $\rho = 0$ to 0 for a warmup period of 50/100 steps for batch sizes 256/128. However, our ablations (see also Appendix B.3) show that: (a) even without warmup, training succeeds, so the zero-success signal does *not* cause instability, and (b) a much less conservative warmup of just 10 steps works as well. Thus, in practice, we recommend a short warmup of 10–20 steps when the base model's initial success rate is very low (<1.5%), which is easy to assess before training.

Figure 4 shows that Plateau-R slightly outperforms GRPO for both batch-sizes, demonstrating the benefit of asymmetric prompt weighting.

This setup follows that in the blog post (Schulman & al, 2025) where the stepsize was tuned for GRPO; and we achieve the same performance with GRPO in our setup.

### 4.3. Post-SFT RL: MATH

Next, we consider the MATH dataset (Hendrycks et al., 2021) consisting of 12k training examples and 500 test examples. Each task is a math problem of varying difficulty levels (1-5) with an answer that is easy to verify, for example a number or fraction. The binary verifier simply compares the solution suggested by the LLM with the provided solution, being lenient with regards to formatting (e.g., \frac{13}{21} is the same as 13/21). MATH is frequently used for evaluating GRPO-style algorithms since it provides a challenging set of problems with an automatic and final-answer checker as reward.

We use the Llama-3.2-3b-instruct model rather than models from the Qwen-series (as common in the literature) because, on MATH, Qwen trained with GRPO-style RLVR improves even with random, incorrect, or format-only rewards (Shao et al., 2025), suggesting the benchmark signal may be confounded by model-specific priors and/or contamination. Thus, using Qwen would make it difficult to attribute performance differences to differences in advantage design. We choose $M = 16$ and our batch-size is 256 (256 distinct math problems and for each $M = 16$ completions).

From the test accuracy (Pass@1) during training for four epochs in Figure 5(left) it can be seen that the performance of all advantage estimators is very similar. Starting from a score of 0.38, all variants achieve scores in the range $0.53 - 56$. See Figure 7 for the distributions of $\hat{\rho}$ during training with Linear-R. For TinyZero, we saw that Linear-R improves over the other advantage estimators in the second half of training by improving on examples with low values of $\rho$. For MATH, we don't see such an improvement which is somewhat expected, considering that at step 90 there are almost no examples with close-to-zero (but not equal to zero) values of $\hat{\rho}$.

### 4.4. Post-SFT RL: DAPO math

Finally, we consider a post-SFT RL setup where we take a model that is already very good at reasoning through supervised finetuning (SFT) on reasoning traces, and improve it further through RL. We choose this setup because models are often SFT'd first and then further improved through RL.

We follow closely the JustRL setup (He et al., 2025) for its simple and clean setup, and so that we can start from well-chosen hyperparameters. We train the DeepSeek-R1-Distill-Qwen-1.5B model, and use the DAPO-math dataset (the english subset) (Yu et al., 2026) for training. The batch size is 256 prompts, and we take $M = 8$ samples per prompt. The learning rate is 1e-6 and we perform on-policy training without KL penalty nor clipping.

Figure 5(right) shows the results: Similar as in Section 4.3 for MATH; all four considered methods perform similarly well, and there is no benefit of up-weighting the difficult prompts (Plateau-R, Linear-R, Sqrt-R), but it also does not hurt performance.

## 5. A policy-dynamics perspective to weighting

Here, we ask: How do different prompt weights $\omega_x$ shape the dynamics of RLVR. To answer this, we study how the success probability evolves over training and how different choices of weight functions $\omega_x$ in Equation (5) induce qualitatively different convergence behaviors. We build on the policy-dynamics analysis of Mroueh (2025), extending it using the general reweighting framework of Section 3 to cover a broad family of policy optimization algorithms.

We find that in a low-success regime an asymmetric prompt weighting (Sqrt-R) is best, and otherwise GRPO is better; which supports our empirical finding that asymmetric prompt weightings are best in from-scratch RL setups that start from a low-success regime.

### 5.1. Theoretical model

To isolate the effect of the weighting $\omega_x$ on the learning dynamics, we consider an idealized, population-level version of policy optimization objectives for RLVR that captures the core ingredients relevant for dynamics: binary rewards, response-level advantages, and a proximal regularization that prevents overly large policy updates. We consider the following policy optimization per prompt $x$ at each iteration $t$ of the algorithm:

$$\max_{\pi_t(\cdot|x)} \mathbb{E}_{y \sim \pi_t(\cdot|x)} A_y(\rho_{t-1}) - \beta D_{\text{KL}}\big(\pi_t(\cdot|x)||\pi_{t-1}(\cdot|x)\big),$$
(7)

where $A_y(\rho) = \omega_x(\rho) \cdot (r(y) - \rho)$ is the advantage and $\omega_x$ the weight function, as throughout (cf. Equation (5)). Here, $\pi_t$ denotes the policy to be optimized at time $t$ and $\pi_{t-1}$ is the policy of the previous iteration. Unlike the rest of the paper, here we consider the population limit over generated responses $M \to \infty$; thus, the advantage $A_y(\rho_{t-1})$ of a response $y$ depends on the (population) success rate $\rho_{t-1}$ of the sampling policy $\pi_{t-1}$ at the previous iteration. $D_{\text{KL}}$ is the KL divergence and $\beta > 0$ a regularization parameter.

Optimizing directly over the policy distribution, rather than over a parametric representation is standard abstraction in the theoretical RL literature, e.g., (Rafailov et al., 2023;

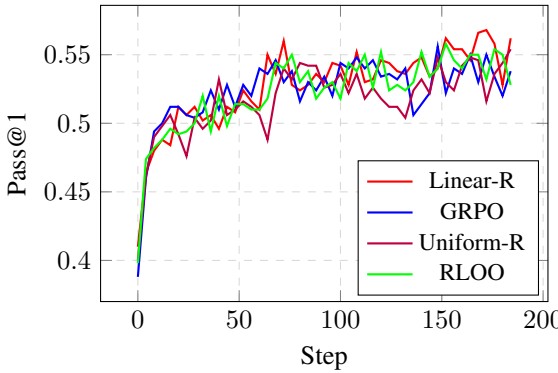 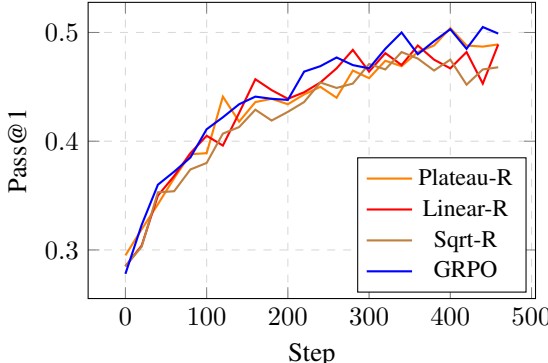

*Figure 5.* Post-SFT RL: Test accuracy (i.e., fraction of correctly solved problems) during RL on the MATH dataset **(left)** and on DAPO math **(right)**. All methods perform similarly well.

Mroueh, 2025; Vojnovic & Yun, 2025; Roux et al., 2025); it allows us to focus on how probability mass shifts between correct and incorrect responses over time.

### 5.2. Success-rate dynamics

As shown in Appendix D.1 by taking the continuous time limit to the solution of (7), the success rate evolves according to the ordinary differential equation (ODE):

$$\frac{\mathrm{d}\rho_t}{\mathrm{d}t} = \frac{1}{\beta}\rho_t(1-\rho_t)\cdot\omega(\rho_t)\,. \tag{8}$$

Thus, the rate of improvement is governed by the uncertainty term $\rho_t(1-\rho_t)$ and the weight $\omega(\rho_t)$. Different weights emphasize different regions of the success probabilities.

In practical RLVR, each unit of optimization time corresponds to drawing a finite number of rollouts per prompt. When the success rate $\rho$ is small, the dominant cost is not captured by the optimization clock $t$, but rather by the number of rollouts required before observing a successful response that yields a positive reward.

Under a Bernoulli outcome model, the probability of observing at least one success in $M$ rollouts is $1-(1-\rho)^M \approx M\rho$ in the low-success regime $\rho \ll 1/M$. Thus, the expected number of rollouts per informative signal scales as $1/(M\rho)$. In this rare-success regime, one unit of population time costs on the order of $g(\rho) = 1/\rho$ rollout units. This motivates measuring progress in an *effective time* variable $\tau$ that upweights time spent at small $\rho$:

$$\mathrm{d}\tau = g(\rho_t)\,\mathrm{d}t, \qquad g(\rho) = 1/\rho. \tag{9}$$

Applying this change of variables to Equation (8), the success rate $\rho_\tau$ evolves according to the effective-time ODE:

$$\frac{\mathrm{d}\rho_\tau}{\mathrm{d}\tau} = \frac{1}{\beta}\,\rho_\tau^2(1-\rho_\tau)\cdot\omega(\rho_\tau). \tag{10}$$

### 5.3. Optimal weighting

Given the dynamics for regular and effective time, we now ask which choice of the weight function $\omega$ leads to the fastest improvement in success probability. We find that for regular time GRPO is optimal, while for effective time Sqrt-R is optimal.

**Regular-time optimality of GRPO.** Fix prompt $x$ and let its average success probability $\rho_t$ at iteration $t$ evolve according to the ODE in Equation (8). Our goal is to minimize the time required for the success rate to reach a target value $\rho_*$ from initial value $\rho_0$. To exclude trivial solutions, and to ensure a fair normalization across all weight functions, we impose a budget $B$ on the total allowed change of the weights in the interval $[\rho_0, \rho_*]$:

$$\int_{\rho_0}^{\rho_*} \omega(\rho)\,\mathrm{d}\rho \leq B\,, \tag{11}$$

where we also consider $\omega(\rho) \geq 0$. Without loss of generality, fix $B = 1$ onwards. Such a budget constraint allows us to fairly compare among algorithms with different weights by normalizing the total "update budget". Specifically, Eq. (11) bounds the cumulative magnitude of the per-prompt weighting coefficient $\omega_x(\rho)$ used in Equation (3). While one could alternatively normalize the optimization using the effective weight $\omega(\rho) \cdot \rho(1-\rho)$ that scales the gradient difference $\widehat{\nabla}_1 - \widehat{\nabla}_0$ (see Equation (4)), this specific alternative normalization does not qualitatively alter the main conclusion of our analysis: regular-time dynamics favor symmetric weightings, whereas effective-time dynamics in the rare-success regime favor asymmetric weightings.

**Proposition 5.1.** *Suppose the success rate $\rho_t$ of a fixed prompt evolves as in Equation (8) with initialization $\rho_0$. Let $T(\rho_0, \rho_*; \omega)$ be the time required so that $\rho_t = \rho_*$ for target success rate $\rho_* > \rho_0$. The non-negative weight $\omega : [0,1] \to \mathbb{R}_{\geq 0}$ that minimizes $T$ subject to the budget constraint (11) with $B = 1$ is: $\omega_{opt}(\rho) \propto \frac{1}{\sqrt{\rho(1-\rho)}}, \quad$ for $\rho \in [\rho_0, \rho_*]$.*

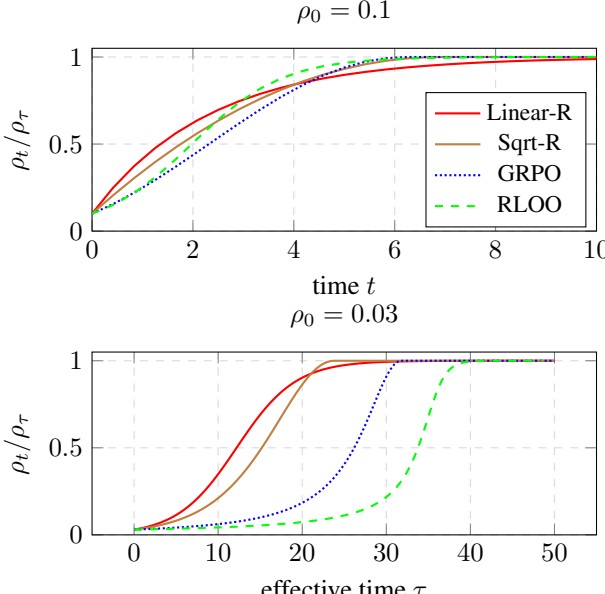

*Figure 6.* Comparison of success-probability dynamics $\rho$ governed by the ODE in Eq. (8) (with $\beta = 1$) for RLOO, GRPO, Linear-R and Sqrt-R, with their weights normalized to satisfy the same total budget constraint Eq. (11) (with $B = 1$). **Left:** regular time $t$ with initialization $\rho_0 = 0.1$. **Right:** effective time $\tau$ with initialization $\rho_0 = 0.03$, where $\tau$ accounts for rollout cost in the low-success regime. As predicted by Propositions 5.1 and 5.2, GRPO is optimal in regular time, whereas Sqrt-R is optimal in effective time in terms of time required to reach target success rate $\rho_* = 1$. The graphs plot the dynamics stated explicitly in Propositions D.1 and D.2, using the budget normalization from Lemma D.3.

Here, time $T$ is the continuous analogue to the number of non-zero gradient updates applied to the model. The proposition gives an optimality interpretation to GRPO's weighting as the weighting minimizing the optimization clock, i.e., the number of updates to reach a target success rate, when all weight functions are normalized to satisfy the same budget constraint.

Figure 6 (left) visualizes this optimality result, showing the evolution of the success rate for Linear-R, Sqrt-R, RLOO, and GRPO with weights normalized to satisfy the budget constraint for $\rho_* = 1$ (see Proposition D.1 for analytical expressions). As predicted, GRPO reaches the target $\rho_* = 1$ first. However, the gain over other algorithms is modest, and asymmetric weights, while not optimal, perform comparably. This is consistent with our Post-SFT experimental findings where all methods achieved similar performance.

**Effective time-optimality of Sqrt-R in low-success regimes.** We now characterize the weight function that minimizes the effective time to reach a target success rate in effective time. As before, for fair comparison across different weights, we normalize by imposing the same budget constraint (11) on the total accumulated weight.

**Proposition 5.2.** *Suppose the success rate $\rho_t$ of a fixed*

*prompt evolves as in Equation (8) with initialization $\rho_0 \in (0, 1)$ and target $\rho_* > \rho_0$. For $g(\rho)$ given in Equation (9), define effective time-to-target as*

$$T_g(\rho_0, \rho_*; \omega) := \int_{\rho_0}^{T} g(\rho_t) \, \mathrm{d}t, \qquad where \; \rho_T = \rho_*.$$

*Under the budget constraint (11) with $B = 1$, the non-negative weight function minimizing $T_g$ over $[\rho_0, \rho_*]$ is*

$$\omega_{\mathrm{opt}}(\rho) \propto \frac{1}{\rho\sqrt{1 - \rho}}. \tag{12}$$

Propositions 5.1 and 5.2 optimize the same population dynamics (8) but under different notions of time. The optimization time $t$ yields the GRPO weighting whereas the rollout-cost-adjusted effective time (9), which more accurately reflects computational costs when $\rho$ is small, yields the weight (12). This is exactly the weight of our Sqrt-R algorithm. Also, it behaves as Linear-R's $1/\rho$ weight in the low-success regime.

Figure 6 (right) visualizes this optimality under budget normalization, showing that Sqrt-R reaches $\rho_* = 1$ significantly faster than all other algorithms in effective time, with Linear-R also substantially outperforming GRPO and RLOO (see Proposition D.2 in the appendix for analytical expressions). This distinction is practically relevant: from-scratch RL often operates for a substantial period in a low-success regime where successful rollouts are scarce, making effective time the more appropriate metric. In these settings, both our theory and experiments yield that Sqrt-R and Linear-R outperform the strong GRPO baseline.

Under the abstractions of optimizing over the policy and population-level objectives, the optimal policy for all weighting schemes approaches $\rho_* = 1$ eventually, albeit faster for asymmetric weights in low success regimes. Relaxing these assumptions (e.g., by incorporating parameterization over model architecture) could also explain the empirical findings of Section 4 on different plateau levels between asymmetric and symmetric weightings.

# 6. Discussion and limitation

We have demonstrated that asymmetric prompt weightings can be beneficial, in particular for from-scratch RLVR. While our analysis focuses on binary rewards, the proposed weightings and advantage estimators extend naturally to rewards in $[0, 1]$ by taking $\hat{\rho}$ to be the empirical mean reward. Several directions merit further study: asymmetric weighting beyond binary rewards; applications to training agentic systems; and scaling to larger models and datasets. Our theory also suggests allocating more rollouts when $\rho$ is close to 0.

## Acknowledgements

We thank the anonymous reviewers for their thoughtful feedback.

RH gratefully acknowledges the computing time made available for the RL experiments in this paper on the high-performance computer at the NHR Center of TU Dresden. This center is jointly supported by the Federal Ministry of Research, Technology and Space of Germany and the state governments participating in the NHR (www.nhr-verein.de/unsere-partner).

The work of MS was partially supported by AWS credits through an Amazon Faculty Research Award, a NAIRR Pilot Award, and generous funding by Coefficient Giving. M. Soltanolkotabi is also supported by the Packard Fellowship in Science and Engineering, a Sloan Research Fellowship in Mathematics, NSF CAREER Award #1846369, NSF CIF Awards #1813877 and #2008443, and NIH Award DP2LM014564-01.

CT gratefully acknowledges support from the Natural Sciences and Engineering Research Council (NSERC) of Canada via Discovery Grant No. 2021-03677 and Alliance Grant ALLRP 581098-22, and a research gift from Google.

## Impact Statement

This paper presents work whose goal is to advance the field of machine learning. There are many potential societal consequences of our work, none of which we feel must be specifically highlighted here.

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

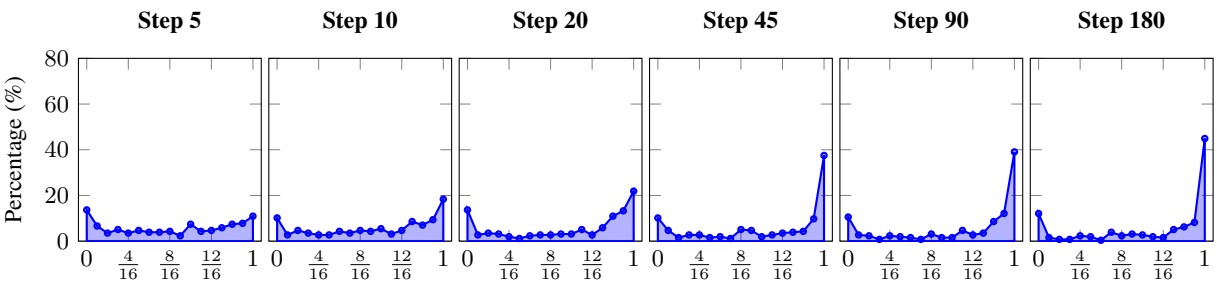

*Figure 7.* Distribution of the fraction of correct completions, $\widehat{\rho}$, out of the 16 completions for each prompt during a run of Linear-R for MATH

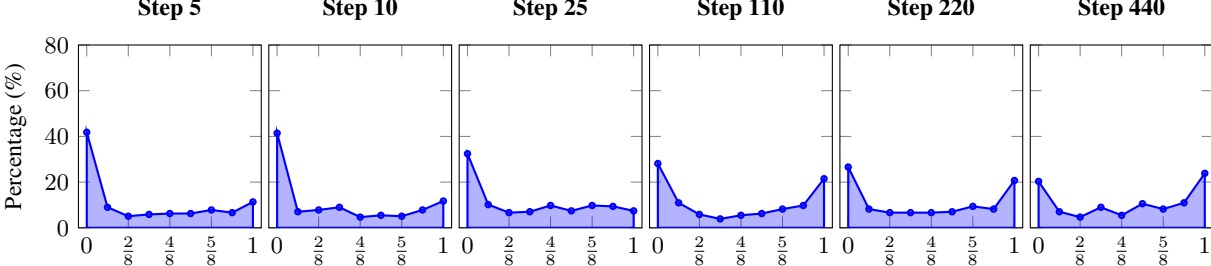

*Figure 8.* Distribution of the fraction of correct completions, $\widehat{\rho}$, out of the 8 completions for each prompt during a run of Linear-R for DAPO math (see Section 4.4).

## A. Supplementary figures and table

Figure 7 and 8 show the distribution of the fraction of correct responses during Linear-R runs on the MATH and DAPO-math experiments discussed in the main body.

Table 1 contains the effective weights for the considered algorithms.

## B. Ablation studies

In this section, we present ablation studies.

### B.1. Importance of assigning non-zero weights to zero-success groups

The asymmetric weightings we consider in this paper assign non-zero weight to prompts with $\widehat{\rho} = 0$. As mentioned this is important for performance. To demonstrate this, we performed further TinyZero experiments for Plateau-R, for which we assign zero weight to zero-success prompts (i.e., groups with $\widehat{\rho} = 0$).

See Figure 9 for the results. It can be seen that with the same stepsize, training is less stable, and leads to worse performance. A smaller stepsize fixes the instability, but results in worse performance.

We also considered the weight $\omega_x = \sqrt{(1-\rho)/\rho}$, which is another example of an asymmetric weight that assigns zero

*Table 1.* Overview of algorithms' *effective* weights multiplying the difference of averaged positive and negative gradients $\widehat{\nabla}_1 - \widehat{\nabla}_0$ for binary rewards.

| | Proposed Weightings | | | | Standard Baselines | |
| Algorithm | Linear-R | Sqrt-R | Plateau-R | Uniform-R | GRPO | RLOO |
|---|---|---|---|---|---|---|
| **Effective Weight** | $1 - \hat{\rho}$ | $\sqrt{1-\hat{\rho}}$ | $\begin{cases} 1/2, & \hat{\rho} < 0.5 \\ \sqrt{\hat{\rho}(1-\hat{\rho})}, & \hat{\rho} \geq 0.5 \end{cases}$ | $1$ | $\sqrt{\hat{\rho}(1-\hat{\rho})}$ | $\hat{\rho}(1-\hat{\rho})$ |

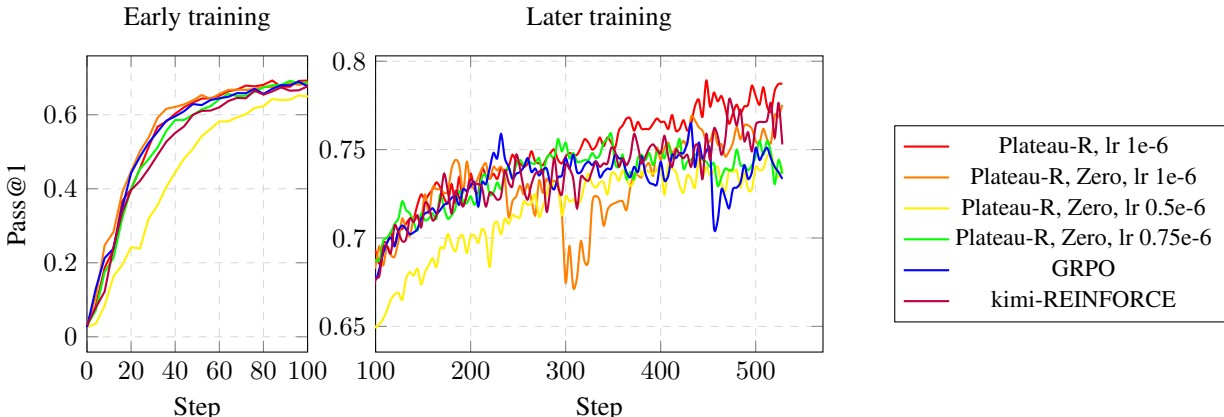

*Figure 9.* From-scratch RL: TinyZero, ablation study on the importance of assigning non-zero weights to zero-success groups. Plateau-R modified with assigning zero weight to zero-success groups ($\hat{\rho} = 0$) performs worse and is less stable; adjusting the learning rate does not fix this. Still, this modification is better than GRPO. Thus, both assigning non-zero weight to prompts with $\rho = 0$ and upweighting low-success prompts are important for performance.

weight to zero-success groups. We call this kimi-weighting since the prioritized sampling strategy used by Team et al. (2025b) where samples are reweighted by $1 - \hat{\rho}$, can be interpreted through surrogate reward maximization yielding that weight as shown by Thrampoulidis et al. (2026).

As seen in Figure 9, kimi-weighting performs worse than the other asymmetric weightings.

### B.2. Run-to-run differences

There are of course run-to-run-differences for our results; to demonstrate that those are relatively modest, we performed three independent runs of Linear-R for TinyZero. The results in Figure 10 demonstrate that the run-to-run differences are small and thus the gains we observe over algorithms with symmetric prompt weightings are consistent.

### B.3. On the warmup of the GSM8K experiment

In Section 4.2, we applied a warmup period during which the advantages for prompts with $\hat{\rho} = 0$ are set to zero. We ablated the length of the warup period and find that the zero-success signal does not cause instability: even without warmup, training succeeds and achieves end-performance that is at least as good as that of GRPO, despite an initial slow phase over the first roughly 80 steps before climbing rapidly. Furthermore, a brief warmup of just 10 steps entirely eliminates this slow phase and leads to performance gains comparable to those reported with the 100-step warmup, which was overly conservative. In practice, we recommend a short warmup of 10–20 steps when the base model's initial success rate is very low ($<1.5\%$), which is easy to assess before training. Intuitively, when $\geq 99\%$ of prompts have $\hat{\rho} = 0$, the zero-success gradient, which pushes away from incorrect completions without a positive directional signal, dominates the update, providing less informative gradients. As more prompts achieve $\hat{\rho} > 0$, e.g., after a short warmup, this zero-success signal becomes a useful complement.

### B.4. Importance of gradient direction

In this work we vary the gradient weights $\omega_x(\rho)$ and choose the gradient direction $\hat{d}_x$ as the direction used by RLOO and GRPO. Choosing the gradient direction as the REINFORCE direction (see Section 2.1, which effectively does not leverage gradients corresponding to zero-rewards) yields significantly worse performance: See Figure 11 for a comparison of Linear-R vs the Linear-R weight with the REINFORCE direction (rejection sampling). We refer to this as rejection sampling as it corresponds to the log-type surrogate advocated in (Davis & Recht, 2025) and their proposed implementation via rejection sampling. See also Section D.7.

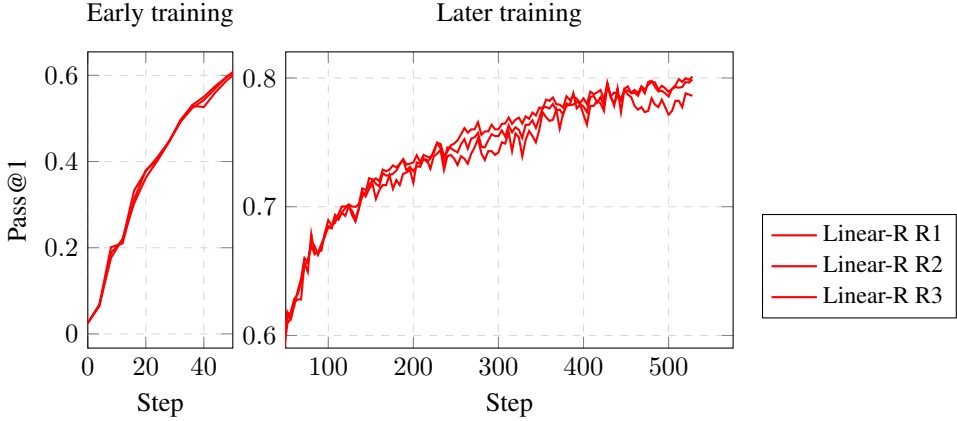

*Figure 10.* From-scratch RL: TinyZero. Test accuracy (i.e., fraction of correctly solved problems, Pass@1) during reinforcement learning. Three independent runs of Linear-R are shown to demonstrate that the run-to-run differences are small and gains are consistent.

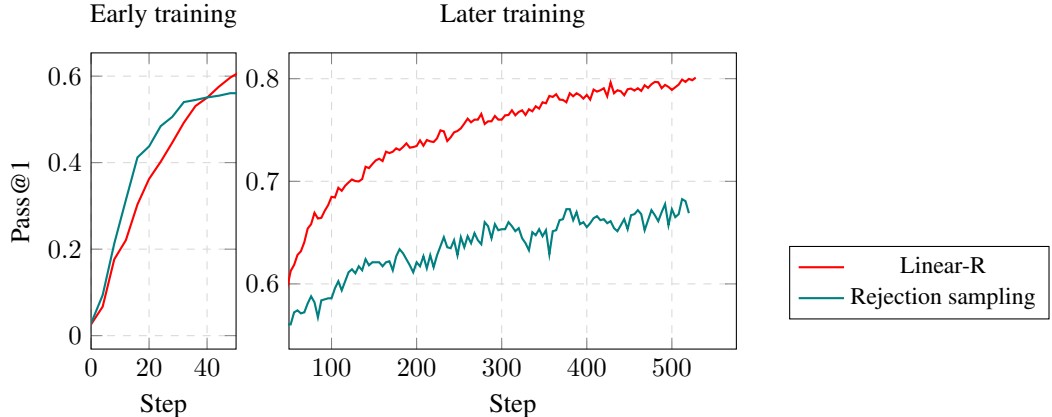

*Figure 11.* From-scratch RL: TinyZero. Test accuracy (i.e., fraction of correctly solved problems, Pass@1) during reinforcement learning. Linear-R vs Rejection sampling,

## C. Experimental details

Here we briefly describe the experimental details. As mentioned, for our RL experiments, we used the SkyRL codebase (Cao et al., 2025), and we have implemented our advantage estimators within that framework; the code is in the supplementary material.

The hyperparameters used in the experiments are in Table 2. We used the same hyperparameters (such as batchsize, etc) within each of the four experimental setups for different advantage estimators, apart from the stepsize, where we considered and used different ones. Specifically we considered (2e-6,1e-6,5e-7) for some experiments for the different advantage estimators.

For the TinyZero experiment the stepsizes for the runs shown are 1e-6 for each advantage estimator but for RLOO, where it is 2e-6. Looking at effective weights in Figure 2 it makes sense that a stepsize that is good for RLOO should be twice of that of GRPO, given that the effective weights are very similar if we multiply the RLOO effective weight with 2. Applying the same thinking to Plateau-R, it might seem reasonable to decrease the stepsize of Plateau-R too relative to the other advantage estimators but here the situation is different: the effective batchsize (corresponding to completions with non-zero weight) for the Plateau-R is larger than for RLOO, and for larger batch sizes the stepsize can often be larger too.

We trained on 1-2 nodes each with 4xH100 and our experiments altogether took 12k H100 hours.

| Hyperparameter | TinyZero | GSM8K | MATH | DAPO Math |
|---|---|---|---|---|
| Use KL Loss | No | No | No | No |
| Train Batch Size | 256 | 128/256 | 128 | 256 |
| Policy Mini Batch Size | 64 | 32/64 | 32 | 64 |
| Micro Train Batch Size / GPU | 8 | 4 | 8 | 4 |
| Micro Forward Batch Size / GPU | 8 | 4 | 8 | 4 |
| Max Prompt Length | 512 | 1024 | 1024 | 1024 |
| Max Response Length | 1024 | 2048 | 3072 | 7168 |
| Update Epochs Per Batch | 1 | 1 | 1 | 1 |
| Learning Rate | 1e-6 | 1e-6 | 5e-7 | 1e-6 |
| Max Grad Norm (Clip) | 0.5 | 1.0 | 1.0 | 1.0 |
| Clip Ratio Range | [0.2, 0.2] | [0.0, 0.0] | [0.2, 0.2] | [0.0, 0.0] |
| Sampling Temperature | 1.0 | 1.0 | 1.0 | 1.0 |
| Responses per prompt $M$ | 16 | 16 | 16 | 8 |

*Table 2.* Hyperparameters for the four sets of RL experiment.

# D. Proofs

### D.1. The continuous limit of success-rate dynamics: Proof of Equation (8)

The optimization problem in (7) is concave. Thus, rewriting its first-order optimality condition gives the following recursion for the optimal policies (see (Mroueh, 2025)):

$$
\pi_t(y|x) = \frac{\pi_{t-1}(y|x) \cdot \exp\left(\frac{1}{\beta} \cdot A_y(\rho_{t-1})\right)}{\mathbb{E}_{y'\sim\pi_{t-1}(\cdot|x)}\pi_{t-1}(y'|x)\exp\left(\frac{1}{\beta} \cdot A_{y'}(\rho_{t-1})\right)} \tag{13}
$$

$$
= \frac{\pi_{t-1}(y|x) \cdot \exp\left(\frac{1}{\beta} \cdot A_y(\rho_{t-1})\right)}{\rho_{t-1}\exp\left(\frac{1}{\beta} \cdot A_1(\rho_{t-1})\right) + (1-\rho_{t-1})\exp\left(\frac{1}{\beta} \cdot A_0(\rho_{t-1})\right)} .
$$

In the second equality, we used that rewards are binary, we recalled the definition

$$
\rho_{t-1} = \mathbb{E}_{y\sim\pi_{t-1}(\cdot|x)}\mathbb{I}\left[r(y) = 1\right] ,
$$

and let $A_{0/1}$ be the advantage scores of wrong/correct responses. This expression makes explicit how the next policy re-weights correct and incorrect responses relative to the previous policy. Importantly, because the update depends on the policy only through its current success rate, the induced dynamics can be described entirely in terms of how this success rate evolves over time. Concretely, using $\rho_t = \mathbb{E}_{y\sim\pi_t(\cdot|x)}\mathbb{I}\left[r(y) = 1\right] = \int \mathrm{d}\pi_t(\cdot|x)\mathbb{I}\left[r(y) = 1\right]$ yields the following recursion over success rates:

$$
\rho_t = \frac{1}{1 + \frac{1-\rho_{t-1}}{\rho_{t-1}}\exp\left(\frac{1}{\beta}\left(A_0(\rho_{t-1}) - A_1(\rho_{t-1})\right)\right)} .
$$

Substituting the general advantage expressions from Eq. (5) for weight function $\omega(\rho)$ we get the following recursion for the population (over model responses) success rate:

$$
\rho_t = \frac{1}{1 + \frac{1-\rho_{t-1}}{\rho_{t-1}}\exp\left(-\frac{1}{\beta}\omega(\rho_{t-1})\right)} .
$$

To further simplify the dynamics, it is convenient to work with the log-odds of the success probability,

$$
L_t := \log\frac{\rho_t}{1 - \rho_t}.
$$

This change of variables linearizes multiplicative updates in probability space and reveals that learning proceeds additively in log-odds space.

$$\frac{1 - \rho_t}{\rho_t} = \frac{1 - \rho_{t-1}}{\rho_{t-1}} \exp\left(-\frac{1}{\beta}\omega(\rho_{t-1})\right) \implies L_t = L_{t-1} + \frac{1}{\beta}\omega(\rho_{t-1}).$$

The log-odds of the success rate increases by a step size proportional to the gradient of the surrogate reward. The derivation up to this point follows (Mroueh, 2025, Thm. 2), extended here to general weight functions $\omega(\rho)$.

To gain further intuition about these dynamics and the role played by the weight, we consider a continuous-time limit of the log-odds update. Concretely, let us consider a continuous-time limit of the form

$$\frac{\mathrm{d}L_t}{\mathrm{d}t} = \frac{1}{\beta}\omega(\rho_t) \implies \frac{\mathrm{d}\rho_t}{\mathrm{d}t} = \frac{1}{\beta}\rho_t(1 - \rho_t)\omega(\rho_t),$$

where for the latter implication we used the chain rule.

### D.2. Dynamics for regular time for RLOO, Linear-R, Sqrt-R, and GRPO

**Proposition D.1.** *Solving the ODE* (8)*, i.e.,*

$$\frac{\mathrm{d}\rho_t}{\mathrm{d}t} = \frac{1}{\beta}\rho_t(1 - \rho_t) \cdot \omega(\rho_t).$$

*for various weights $\omega$ and for initialization $\rho_0 \in (0, 1)$ gives:*

- *RLOO, $\omega(\rho) = 1$: $\rho_t = \left(1 + \left(\frac{1 - \rho_0}{\rho_0}\right)e^{-t/\beta}\right)^{-1}$*

- *Linear-R, $\omega(\rho) = \frac{1}{\rho}$: $\rho_t = 1 - (1 - \rho_0)e^{-t/\beta}$*

- *Sqrt-R, $\omega(\rho) = \frac{1}{\rho\sqrt{1-\rho}}$: $\rho_t = 1 - \left(\max\left\{\sqrt{1 - \rho_0} - \frac{t}{2\beta}, 0\right\}\right)^2$*

- *GRPO, $\omega(\rho) = \frac{1}{\sqrt{\rho(1-\rho)}}$: $\rho_t = \sin^2\left(\min\left(\frac{\pi}{2}, \frac{t}{2\beta} + \arcsin(\sqrt{\rho_0})\right)\right)$*

*Proof.* We prove each case separately.

**RLOO:** For RLOO, the ODE becomes

$$\frac{\mathrm{d}\rho}{\mathrm{d}t} = \frac{1}{\beta}\rho(1 - \rho),$$

and integration yields

$$\int \frac{1}{\rho(1 - \rho)}d\rho = \int \frac{1}{\beta}dt.$$

Solving this gives

$$\log\rho - \log(1 - \rho) - (\log\rho_0 - \log(1 - \rho_0)) = \frac{t}{\beta},$$

which after some algebra gives the stated solution for RLOO above.

**Linear-R:** Next consider Linear-R, for which the ODE becomes

$$\frac{\mathrm{d}\rho}{\mathrm{d}t} = \frac{1}{\beta}(1 - \rho).$$

This is a linear ODE

$$\frac{d}{dt}(1 - \rho) = -\frac{1}{\beta}(1 - \rho),$$

with solution

$$(1 - \rho) = (1 - \rho_0)e^{-t/\beta} \, .$$

which after re-arranging yields the expression for Linear-R.

**Sqrt-R:** For Sqrt-R, the ODE becomes

$$\frac{d\rho}{dt} = \frac{1}{\beta}\sqrt{1 - \rho}$$

Let $u_t := 1 - \rho_t$. Then, while $u_t > 0$:

$$\frac{du_t}{dt} = -\frac{1}{\beta}\sqrt{u_t} \implies 2\sqrt{u_t} = 2\sqrt{u_0} - \frac{1}{\beta}t \, .$$

Therefore $u_t = \left(\max\{\sqrt{u_0} - \frac{t}{2\beta}, 0\}\right)^2$ which finishes the proof.

**GRPO:** For GRPO, the ODE becomes

$$\frac{d\rho}{dt} = \frac{1}{\beta}\sqrt{\rho(1 - \rho)}.$$

Let $\theta_t = \arcsin(\sqrt{\rho_t})$. Then $\sin^2(\theta) = \rho$, and by the chain rule,

$$\frac{d\rho}{dt} = \frac{d}{dt}\left(\sin^2\theta\right) = 2\sin\theta\cos\theta\,\frac{d\theta}{dt} = 2\sqrt{\rho}\sqrt{1 - \rho}\,\frac{d\theta}{dt}$$

Equating this with the ODE gives

$$2\sqrt{\rho(1 - \rho)}\,\frac{d\theta}{dt} = \frac{1}{\beta}\sqrt{\rho(1 - \rho)}.$$

Thus, for $\rho_t \in (0, 1)$ we obtain the linear ODE

$$\frac{d\theta}{dt} = \frac{1}{2\beta}.$$

Integrating and using $\theta_{t=0} = \arcsin(\sqrt{\rho_0})$ yields

$$\theta(t) = \frac{t}{2\beta} + \arcsin(\sqrt{\rho_0}).$$

This derivation is valid as long as $\rho_t < 1$, which corresponds to $\theta_t < \pi/2$. Once $\theta_t$ reaches $\pi/2$, $\rho_t$ reaches 1. Since $\rho = 1$ is an equilibrium point, the solution stays at 1 thereafter. Therefore, the general solution is

$$\rho_t = \sin^2\left(\min\left(\frac{\pi}{2}, \frac{t}{2\beta} + \arcsin(\sqrt{\rho_0})\right)\right),$$

which concludes the proof for GRPO. $\qquad\square$

### D.3. Effective time dynamics for Linear-R, Sqrt-R, RLOO, and GRPO

**Proposition D.2.** *Solving the effective-time ODE* (10), *i.e.,*

$$\frac{d\rho_\tau}{d\tau} = \frac{1}{\beta}\rho_\tau^2(1 - \rho_\tau) \cdot \omega(\rho_\tau)$$

*for various weights $\omega$ and for initialization $\rho_{\tau=0} = \rho_0 \in (0, 1)$ gives:*

- *Linear-R, $\omega(\rho) = \frac{1}{\rho}$: $\rho_\tau = \frac{1}{1 + \left(\frac{1 - \rho_0}{\rho_0}\right)e^{-\tau/\beta}}$*

- *Sqrt-R,* $\omega(\rho) = \frac{1}{\rho\sqrt{1-\rho}}$: *Define* $\tau_* := 2\beta \operatorname{arctanh}\left(\sqrt{1-\rho_0}\right)$. *Then,*

$$\rho_\tau = \begin{cases} \operatorname{sech}^2\left(\operatorname{arctanh}\left(\sqrt{1-\rho_0}\right) - \frac{\tau}{2\beta}\right), & \tau \le \tau_\star, \\ 1, & \tau > \tau_\star. \end{cases}$$

- *GRPO,* $\omega(\rho) = \frac{1}{\sqrt{\rho(1-\rho)}}$: *Define* $z_0 := \sqrt{\frac{1-\rho_0}{\rho_0}}$ *and* $\tau_* := 2\beta z_0$. *Then,*

$$\rho_\tau = \frac{1}{1 + \left(\max\left\{z_0 - \frac{\tau}{2\beta}, 0\right\}\right)^2}$$

  *In particular,* $\rho_\tau = 1$ *for all* $\tau \ge \tau_*$.

- *RLOO,* $\omega(\rho) = 1$: *Define* $s_\tau := \frac{\tau}{\beta} - \frac{1}{\rho_0} + \log\left(\frac{\rho_0}{1-\rho_0}\right)$. *Then* $\rho_\tau$ *is uniquely determined by the implicit relation*

$$-\frac{1}{\rho_\tau} + \log\left(\frac{\rho_\tau}{1-\rho_\tau}\right) = s_\tau,$$

  *and admits an explicit form in terms of the principal branch of the Lambert-W function:*

$$\rho_\tau = \frac{1}{1 + W\left(\exp\left(-s_\tau - 1\right)\right)}.$$

*Proof.* We prove each case separately.

**Linear-R:** $\omega(\rho) = 1/\rho$. The ODE becomes

$$\frac{d\rho}{d\tau} = \frac{1}{\beta}\rho(1-\rho).$$

Separate variables:

$$\frac{d\rho}{\rho(1-\rho)} = \frac{d\tau}{\beta}.$$

Using $\frac{1}{\rho(1-\rho)} = \frac{1}{\rho} + \frac{1}{1-\rho}$, integrate to get

$$\log\left(\frac{\rho}{1-\rho}\right) = \frac{\tau}{\beta} + C.$$

Apply $\rho(0) = \rho_0$ to obtain $C = \log\left(\frac{\rho_0}{1-\rho_0}\right)$. Exponentiating,

$$\frac{\rho}{1-\rho} = \frac{\rho_0}{1-\rho_0}e^{\tau/\beta}.$$

Solving this for $\rho$ gives the result.

**Sqrt-R:** $\omega(\rho) = \frac{1}{\rho\sqrt{(1-\rho)}}$. We solve the ODE

$$\frac{d\rho}{d\tau} = \frac{1}{\beta}\,\rho\sqrt{1-\rho}.$$

Separate variables:

$$\frac{d\rho}{\rho\sqrt{1-\rho}} = \frac{1}{\beta}\,d\tau.$$

Let $u = \sqrt{1-\rho}$, so $\rho = 1 - u^2$ and $d\rho = -2u\,du$. Then

$$\int \frac{d\rho}{\rho\sqrt{1-\rho}} = \int \frac{-2u\,du}{(1-u^2)u} = -2\int \frac{du}{1-u^2} = -2\operatorname{arctanh}(u).$$

Hence

$$\operatorname{arctanh}\left(\sqrt{1-\rho_\tau}\right) = C - \frac{\tau}{2\beta},$$

as long as $C \geq \frac{\tau}{2\beta}$. Applying $\tanh$ and squaring yields

$$\sqrt{1 - \rho_\tau} = \tanh\left(C - \frac{\tau}{2\beta}\right) \implies \rho_\tau = 1 - \tanh^2\left(C - \frac{\tau}{2\beta}\right) = \operatorname{sech}^2\left(C - \frac{\tau}{2\beta}\right).$$

Since $\rho_{\tau=0} = \rho_0 \in (0, 1)$, then $C = \operatorname{arctanh}\left(\sqrt{1 - \rho_0}\right)$, so the solution is

$$\rho_\tau = \operatorname{sech}^2\left(\operatorname{arctanh}\left(\sqrt{1 - \rho_0}\right) - \frac{\tau}{2\beta}\right),$$

as long as $\tau \leq \tau_* = 2\beta \operatorname{arctanh}\left(\sqrt{1 - \rho_0}\right)$. At $\tau = \tau_*$ we have $\rho(\tau_*) = 1$. Since the right-hand side of the original ODE vanishes at $\rho = 1$ (because $1 - \rho = 0$), the constant function $\rho_\tau = 1$ is a solution for all $\tau \geq \tau_*$.

**GRPO:** $\omega(\rho) = 1/\sqrt{\rho(1 - \rho)}$. The ODE becomes

$$\frac{d\rho}{d\tau} = \frac{1}{\beta}\rho^{3/2}\sqrt{1 - \rho}. \tag{14}$$

Define

$$z := \sqrt{\frac{1 - \rho}{\rho}}, \qquad \text{so that} \qquad \rho = \frac{1}{1 + z^2}.$$

Differentiating the latter gives

$$\frac{d\rho}{d\tau} = -\frac{2z}{(1 + z^2)^2}\frac{dz}{d\tau}.$$

Using Eq. (14) and the identity $\rho^{3/2}\sqrt{1 - \rho} = \frac{z}{(1+z^2)^2}$, we get for $z > 0$ the following linear ODE:

$$\frac{dz}{d\tau} = -\frac{1}{2\beta}.$$

Hence

$$z(\tau) = z_0 - \frac{\tau}{2\beta}, \qquad \text{where } z_0 = \sqrt{\frac{1 - \rho_0}{\rho_0}}.$$

This formula is valid as long as $z(\tau) \geq 0$, i.e. for $\tau \leq \tau_* := 2\beta z_0$. At $\tau = \tau_*$ we have $z(\tau_*) = 0$, hence $\rho(\tau_*) = 1$. Since the right-hand side of the original ODE vanishes at $\rho = 1$ (because $1 - \rho = 0$), the constant function $\rho_\tau \equiv 1$ is a solution for all $\tau \geq \tau_*$. By uniqueness of solutions for the $z$-equation on the region $z > 0$ and the fact that $\rho = 1$ is an equilibrium of the $\rho$-equation, the global solution can be written as stated in the lemma.

**RLOO:** $\omega(\rho) = 1$. The ODE becomes

$$\frac{d\rho}{d\tau} = \frac{1}{\beta}\rho^2(1 - \rho).$$

Separate variables:

$$\frac{d\rho}{\rho^2(1 - \rho)} = \frac{d\tau}{\beta}.$$

Using $\frac{1}{\rho^2(1-\rho)} = \frac{1}{\rho^2} + \frac{1}{\rho} + \frac{1}{1-\rho}$ and integrating gives

$$-\frac{1}{\rho} + \log\left(\frac{\rho}{1 - \rho}\right) = \frac{\tau}{\beta} + C,$$

where, by applying initialization $\rho_{\tau=0} = \rho_0$,

$$C = -\frac{1}{\rho_0} + \log\left(\frac{\rho_0}{1 - \rho_0}\right).$$

Thus, $\rho_\tau$ is determined implicitly by solving

$$-\frac{1}{\rho_\tau} + \log\left(\frac{\rho_\tau}{1 - \rho_\tau}\right) = \frac{\tau}{\beta} - \frac{1}{\rho_0} + \log\left(\frac{\rho_0}{1 - \rho_0}\right) =: s_\tau.$$

To see uniqueness, note that the left-hand side is a continuous strictly increasing function of $\rho \in (0, 1)$.

For an explicit form, set $u_\tau := \frac{1-\rho_\tau}{\rho_\tau}$ so that $\rho_\tau = \frac{1}{1+u_\tau}$. Then, the implicit equation becomes

$$-(1 + u) - \log u = s_\tau \iff u + \log u = -(s_\tau + 1).$$

Exponentiating yields

$$ue^u = e^{-(s_\tau+1)}.$$

Therefore

$$u_\tau = W(e^{-(s_\tau+1)}),$$

where $W$ is the principal branch of the Lambert-$W$ function, completing the proof. $\qquad\square$

### D.4. Calculating the constants for budget constraints

In this section, we clarify the constant for a normalization budget on the total weight magnitude applied throughout the learning process that are used to draw Figure 6. To plot the normalized dynamics, we use the closed-form solutions from Propositions D.1 and D.2 with appropriately rescaled $\beta$ values. This works because normalizing $\omega(\rho)$ by a constant $c$ in the ODE $\frac{d\rho}{dt} = \frac{1}{\beta}\rho(1-\rho)\omega(\rho)$ is equivalent to using the unnormalized weight $\omega$ with $\beta$ replaced by an effective $\beta_{\text{eff}} = \beta/c$. For budget-normalized plotting with $\beta = 1$, the effective values are: $\beta_{GRPO} = \pi - 2\arcsin(\sqrt{\rho_0})$, $\beta_{Sqrt-R} = 2\operatorname{arctanh}(\sqrt{1-\rho_0})$, $\beta_{Linear-R} = \ln(1/\rho_0)$, and $\beta_{RLOO} = 1 - \rho_0$.

**Lemma D.3.** *Consider the dynamics in Eq. (8) with initialization $\rho_0 \in (0, 1)$ and target $\rho_* = 1$. Under the budget constraint $\int_{\rho_0}^1 \omega(\rho) \, d\rho = 1$, the normalized weight functions for the algorithms are:*

- *GRPO :*

$$\omega(\rho) = \frac{1}{\pi - 2\arcsin(\sqrt{\rho_0})} \cdot \frac{1}{\sqrt{\rho(1-\rho)}}$$

- *Sqrt-R:*

$$\omega(\rho) = \frac{1}{2\operatorname{arctanh}(\sqrt{1-\rho_0})} \cdot \frac{1}{\rho\sqrt{1-\rho}},$$

- *Linear-R:*

$$\omega(\rho) = \frac{1}{\ln(1/\rho_0)} \cdot \frac{1}{\rho}.$$

- *RLOO:*

$$\omega(\rho) = \frac{1}{1 - \rho_0}.$$

*Proof.* **GRPO:** The raw weight is $w(\rho) = \frac{1}{\sqrt{\rho(1-\rho)}}$. The normalization constant $C$ is determined by:

$$\frac{1}{C} = \int_{\rho_0}^1 \frac{d\rho}{\sqrt{\rho(1-\rho)}} = [2\arcsin(\sqrt{\rho})]_{\rho_0}^1 = \pi - 2\arcsin(\sqrt{\rho_0}).$$

**Sqrt-R:** The raw weight is $w(\rho) = \frac{1}{\rho\sqrt{1-\rho}}$. The normalization constant $C$ is determined by:

$$\frac{1}{C} = \int_{\rho_0}^1 \frac{d\rho}{\rho\sqrt{1-\rho}}.$$

Using the substitution $u = \sqrt{1-\rho}$, we have $d\rho = -2u\,du$ and $\rho = 1 - u^2$. Thus,

$$\int_{\sqrt{1-\rho_0}}^0 \frac{-2u\,du}{(1-u^2)u} = -2\int_{\sqrt{1-\rho_0}}^0 \frac{du}{1-u^2} = -[2\operatorname{arctanh}(u)]_{\sqrt{1-\rho_0}}^0 = 2\operatorname{arctanh}(\sqrt{1-\rho_0}).$$

**Linear-R:** The raw weight is $w(\rho) = 1/\rho$. The normalization constant is $1/C = \int_{\rho_0}^1 \frac{d\rho}{\rho} = -\log(\rho_0)$.

**RLOO:** The raw weight is $w(\rho) = 1$. The normalization constant is $1/C = 1 - \rho_0$. $\qquad\square$

## D.5. Proof of Proposition 5.1

The time required to reach $\rho_t = \rho_*$ starting from $\rho_0$ is

$$T(\rho_0, \rho_*; \omega) = \int_0^T dt = \int_{\rho_0}^{\rho_*} \frac{d\rho}{d\rho_t/dt} . \tag{15}$$

Thus, using the dynamics Eq. (8), the optimization problem becomes

$$\min_{\omega:[0,1]\to\mathbb{R}_{\geq 0}} \int_{\rho_0}^{\rho_*} \frac{d\rho}{\rho(1-\rho)\cdot\omega(\rho)} \quad \text{s.t.} \quad \int_{\rho_0}^{\rho_*} \omega(\rho)\, d\rho \leq 1 .$$

Define $a(\rho) = \frac{1}{\rho(1-\rho)}$. By Cauchy–Schwarz,

$$\left( \int_{\rho_0}^{\rho_*} \sqrt{a(\rho)}\, d\rho \right)^2 \leq \left( \int_{\rho_0}^{\rho_*} \frac{a(\rho)}{\omega(\rho)}\, d\rho \right) \left( \int_{\rho_0}^{\rho_*} \omega(\rho)\, d\rho \right) .$$

Since the second factor is at most 1, we obtain the lower bound $T(\rho_0, \rho_*; \omega) \geq \beta \left( \int_{\rho_0}^{\rho_*} \frac{d\rho}{\sqrt{\rho(1-\rho)}} \right)^2$, with equality if and only if $\omega(\rho) \propto \sqrt{a(\rho)}$ and the budget constraint is tight. Therefore, the optimal choice is

$$\omega_{\mathrm{opt}}(\rho) = c\, \frac{1}{\sqrt{\rho(1-\rho)}} \quad \text{for } \rho \in [\rho_0, \rho_*] ,$$

with $c$ chosen to saturate the budget normalization constraint. Concretely, this gives

$$\omega_{\mathrm{opt}}(\rho) = \frac{1}{2\left( \arcsin\sqrt{\rho_*} - \arcsin\sqrt{\rho_0} \right)} \cdot \frac{1}{\sqrt{\rho(1-\rho)}} .$$

## D.6. Proof of Proposition 5.2

Recall the population dynamics in Eq. (8) and the effective time-to-target definition:

$$T_g(\rho_0, \rho_*; \omega) := \int_{\rho_0}^T g(\rho_t)\, dt, \qquad \text{where } \rho_T = \rho_*,$$

with $g(\rho) = 1/\rho$. Using the change of variables $dt = d\rho / \frac{d\rho_t}{dt}$ yields

$$T_g(\rho_0, \rho_*; \omega) = \int_{\rho_0}^{\rho_*} \frac{g(\rho)}{d\rho_t/dt}\, d\rho = \int_{\rho_0}^{\rho_*} \frac{g(\rho)}{\frac{1}{\beta}\rho(1-\rho)\omega(\rho)}\, d\rho = \beta \int_{\rho_0}^{\rho_*} \frac{d\rho}{\rho^2(1-\rho)\,\omega(\rho)}. \tag{16}$$

Thus, the optimization of minimizing $T_g$ subject to $\omega(\rho) \geq 0$ and the budget constraint $\int_{\rho_0}^{\rho_*} \omega(\rho)\, d\rho \leq 1$ becomes

$$\min_{\omega:[0,1]\to\mathbb{R}_{\geq 0}} \int_{\rho_0}^{\rho_*} \frac{a(\rho)}{\omega(\rho)}\, d\rho \quad \text{s.t.} \quad \int_{\rho_0}^{\rho_*} \omega(\rho)\, d\rho \leq 1.$$

where we defined $a(\rho) := \frac{1}{\rho^2(1-\rho)}$. By Cauchy–Schwarz,

$$\left( \int_{\rho_0}^{\rho_*} \sqrt{a(\rho)}\, d\rho \right)^2 \leq \left( \int_{\rho_0}^{\rho_*} \frac{a(\rho)}{\omega(\rho)}\, d\rho \right) \left( \int_{\rho_0}^{\rho_*} \omega(\rho)\, d\rho \right) .$$

Since the second factor is at most 1, we obtain the lower bound

$$T_g(\rho_0, \rho_*; \omega) \geq \beta \left( \int_{\rho_0}^{\rho_*} \sqrt{\frac{1}{\rho^2(1-\rho)}}\, d\rho \right)^2 ,$$

with equality if and only if

$$\omega(\rho) \propto \sqrt{a(\rho)} = \frac{1}{\rho\sqrt{1-\rho}} \quad \text{for } \rho \in [\rho_0, \rho_*]$$

with the proportionality constant determined to saturate the budget constraint.

### D.7. Origins of Linear-R

We arrive at Linear-R by selecting asymmetric weighting $\omega(\rho) = 1/\rho$ in the general form of RLVR algorithms in Equation (3). We specifically choose such a weight that aggressively focuses on small $\rho$ to encourage larger gradient signal.

We can alternatively arrive at the same weighting by applying the forward-engineering recipe (Thrampoulidis et al., 2026, Sec. 5.3) to a logarithmic surrogate reward. Concretely, start from a surrogate $F(u) = \log(u)$ and optimize $F(\rho_x(\theta))$: direct differentiation yields $F'(\rho_x(\theta)) \cdot \nabla_\theta \rho_x(\theta) = \frac{1}{\rho_x(\theta)} \cdot \nabla_\theta \rho_x(\theta)$, which coincides with our Linear-R update after using an RLOO estimator for $\nabla_\theta \rho_x(\theta)$.

It is worth noting that another popular RL algorithm, which is different to our Linear-R, has also recently been shown to map to the logarithmic surrogate reward by Davis & Recht (2025). In our notation, rejection sampling updates the parameters in the direction

$$\frac{1}{M^+} \sum_{i=1}^{M^+} \nabla_\theta \log \pi_\theta(y|x) = \widehat{\nabla}_1 \tag{17}$$

In population limit, this equals

$$\mathbb{E}_{y \sim \pi_\theta(\cdot|x)}[\nabla_\theta \log \pi_\theta(y|x)|r(y) = 1] = \frac{\mathbb{E}[r(y)\nabla_\theta \log \pi_\theta(y|x)]}{\rho_\theta} = \frac{1}{\rho_\theta} \nabla_\theta \rho_\theta = \frac{\mathrm{d} \log \rho_\theta}{\mathrm{d}\rho_\theta} \,,$$

which corresponds to asymptotically ($M \gg 1$) maximizing the log surrogate reward.

This can also be seen as a special instance of Equation (3) with $\omega_x(\rho) = 1/\rho$ and $\hat{d}_x(\theta) = \widehat{\nabla}_1$. That is, rejection sampling uses a $1/\rho$ weight as Linear-R, but uses the REINFORCE estimate for the gradient rather than the RLOO used for Linear-R.

We also run this rejection sampling weighting for the TinyZero experiment, and it performed significantly worse than the other considered algorithms (Plateau-R, Linear-R, Uniform-R, GRPO, RLOO).

For completeness, we mention that our Sqrt-R can be analogously interpreted as optimizing the surrogate reward $F(u) = -2 \operatorname{arctanh}(\sqrt{1-u})$ using again the RLOO gradient estimator. To see this, recall from Sec. 3 that $F'(\rho) = \frac{1}{\rho\sqrt{1-\rho}}$, which is exactly Sqrt-R's weight.

