# OpenReview forum: "Asymmetric Prompt Weighting for Reinforcement Learning with Verifiable Rewards"
_ICML.cc/2026/Conference — ICML 2026 regular_

### Official Review · Reviewer_HZCE · 2026-03-09

**Soundness:** 3
**Presentation:** 3
**Significance:** 3
**Originality:** 3
**Overall Recommendation:** 4
**Confidence:** 4

**Summary:**

This paper proposes asymmetric prompt-level weightings for RLVR with binary rewards, where prompts with low success probability receive higher gradient weights compared to the symmetric weighting used by GRPO and RLOO. Three variants are introduced: Log-REINFORCE (LR, ω=1/ρ), Asymmetric-REINFORCE (AR), and Cancel-REINFORCE (CR). Experiments on four setups show that LR and AR improve over GRPO in from-scratch RL settings (TinyZero, GSM8K) by ~6%, but show no difference in post-SFT RL settings (MATH, DAPO-math). A theoretical analysis under a population-limit ODE model shows that GRPO is optimal in regular time while LR is asymptotically optimal in effective (rollout-cost-adjusted) time.

**Compliance With Llm Reviewing Policy:**

Affirmed.

**Final Justification:**

The rebuttal addressed my main concerns. The warmup ablations and additional experiments (Hyperphantasia, K&K) strengthen the empirical story. The scope remains somewhat narrow given the method primarily benefits from-scratch low-success regimes, but the paper is clearly written and honest about its limitations. I raise my score from 3 to 4.

**Key Questions For Authors:**

Questions:

1. In TinyZero, the $\\hat{\\rho}=0$ ablation hurts performance, yet in GSM8K the same signal must be disabled for 50-100 warmup steps. What specific property of the task or model determines whether the $\\hat{\\rho}=0$ signal is helpful or harmful? In practice, should we monitor the $\\hat{\\rho}=0$ rate during training and manually decide when to enable or disable this signal?

2. In TinyZero (Figure 2), all algorithms perform similarly during steps 0-200 when $\\hat{\\rho}=0$ prompts are most prevalent. For instance, at step 25 (Figure 3), a large fraction of prompts have $\\hat{\\rho} < 0.5$ — precisely the regime where LR/AR weights diverge most from GRPO (Figure 1) — yet no performance difference is visible. If asymmetric weighting is the key mechanism, why does it not manifest during this phase?

3. The $\\hat{\\rho}$​ distributions in Figure 3 (TinyZero, Step 220) and Figure 8 (MATH, Step 90) appear qualitatively similar, yet LR/AR show gains in the former but not the latter. Could the authors clarify what specific distributional difference they believe explains this discrepancy?

4. Comparing the advantage magnitudes across methods at low $\\hat{\\rho}$​ (e.g., at $\\hat{\\rho}=1/8$, LR assigns +7/-1, AR assigns +4/-0.57, while GRPO assigns +2.65/-0.38), the proposed methods can be viewed as scaling up gradient magnitudes relative to GRPO while using the same learning rate. Have the authors tried comparing against GRPO with a larger learning rate (e.g., 5e-6 instead of 1e-6) to disentangle the effect of asymmetric weighting from simply taking larger gradient steps?

**Limitations:**

The authors acknowledge that their method is limited to binary rewards and that post-SFT settings show no benefit. However, they do not adequately discuss why the $\\hat{\\rho}=0$ signal requires warmup in some tasks but not others, which is a practical limitation that users would need to navigate.

**Strengths And Weaknesses:**

Strengths:

1. The paper provides a clean unified framework that decomposes existing RLVR algorithms (REINFORCE, RLOO, GRPO) into a shared gradient direction $(\\hat{\\nabla}\_1 - \\hat{\\nabla}\_0)$ and a prompt-level weight $\\omega(\\rho)$. This perspective makes explicit how different algorithms prioritize prompts of varying difficulty, and the observation that symmetric weighting may underserve hard prompts is a relevant and well-motivated problem for the RLVR community.
2. The paper evaluates the proposed methods across a diverse set of scenarios, including both from-scratch RL (TinyZero, GSM8K) and post-SFT RL (MATH, DAPO-math) settings. Testing on these varied regimes and transparently reporting the lack of significant gains in the post-SFT setting is commendable.

Weaknesses:
1. A key contribution of this paper is that LR and AR provide a non-zero gradient signal even when all completions fail ($\\hat{\\rho}=0$). However, in TinyZero (Section 4.1), the authors report that zeroing out this signal hurts performance, while in GSM8K (Section 4.2), the same signal "dominates the gradient" and must be disabled via a warmup heuristic for 50-100 steps. Such an ad-hoc heuristic raises concerns about whether this signal must be manually toggled based on the task's success rate. Moreover, even within TinyZero, if $\\hat{\\rho}=0$ gradient signal is the key mechanism, one would expect LR/AR to outperform baselines earliest in training when such prompts dominate (>80% at step 5, Figure 3). Yet Figure 2 shows all algorithms perform similarly during steps 0-200; the gains only emerge later when most prompts already have high $\\hat{\\rho}$​.
2. The SFT-then-RL pipeline is one of the most widely adopted paradigm in LLM training. But AR and LR show no significant improvement over baselines in post-SFT environments (MATH, DAPO-math). While the authors attempt to explain this limitation by pointing to the lack of low-$\\rho$ prompts, their justification is not fully convincing. For instance, the prompt success distribution at Step 220 in Figure 3 (where LR and AR begin to show steep gains) appears remarkably similar to the distribution at Step 90 in Figure 8 (where no such gains are observed).

---

> ### Author Rebuttal · Authors · 2026-03-31
>
> Thank you for the careful read and thoughtful questions. We address each point below.
>
> **Q1. $\hat{\rho}=0$ signal is helpful or harmful? should we monitor the $\hat{\rho}=0$ rate during training?** As shown in our new ablation, the zero-success signal is neither harmful nor unstable in either setting. Even on GSM8K without any warmup, training succeeds but with a slower initial phase (~80 steps). The difference is quantitative, not qualitative: GSM8K's base model starts at 1.3% accuracy, so the vast majority of prompts have $\rho=0$ and the zero-success gradient temporarily dominates; TinyZero starts at 2.5%, where enough prompts provide a positive signal from the start. A short warmup of 10 steps (not 50-100 as in our conservative original setup) is sufficient when starting from very low accuracy. *No monitoring during training is needed in practice; one can simply evaluate the base model's initial accuracy and apply a brief warmup if it is below around 2–3%.* Our new Hyperphantasia experiment (see response to Reviewer EXiS), which starts from a low-success regime, also requires no warmup, further confirming that this is a minor practical consideration rather than a fundamental limitation.
>
> **Q2.  If asymmetric weighting is the key mechanism, why does it not manifest during this phase?** We address this in two parts. First, at early steps (e.g., steps 5 and 25 in Figure 3), while there is substantial mass at $\rho\approx 0$, there is also non-negligible mass on medium-difficulty prompts. The model naturally solves these first before tackling the harder prompts where LR's larger weights have the most impact, which we believe explains the delayed separation. Second, since submission, we have introduced a new algorithm, Sqrt-R, with effective weight $\omega(\rho) = 1/(\rho\sqrt{1−ρ}),$ which combines the best of both worlds: it matches LR's weighting at small ρ and GRPO's at large ρ. Sqrt-R already surpasses GRPO after step 50 on TinyZero, consistent with the mechanism taking effect once medium-difficulty prompts are resolved. We also note that on GSM8K, gains from LR/AR are visible from very early epochs. We will add Sqrt-R in the revision and explain the above mechanisms as prompted by your question.
>
> **Q3. $On \hat{\rho}$​ distributions.** First, in Figure 8 (MATH, Step 90), there remains non-negligible mass in the $\rho\in[0.3,0.6]$ range that GRPO's weighting favors, unlike in Figure 3 (TinyZero, Step 220). Second, while both distributions show substantial mass at $\rho\approx 0$, this mass behaves differently: in Figure 8, it persists essentially unchanged through Step 180, indicating an irreducible difficulty floor beyond the model's reasoning capacity. In contrast, the mass at $\rho\approx 0$ in Figure 3 continues to decrease at later stages, meaning LR/AR's heavier weighting on these hard prompts translates into actual learning signal. In short, LR/AR's advantage materializes only when the hard prompts are difficult but learnable.
>
> **Q4. proposed methods can be viewed as scaling up gradient magnitudes relative to GRPO?** The key insight is that what matters is not the absolute magnitude of the advantage but the relative weighting across different ρ values within the same algorithm. Simply increasing GRPO's learning rate would uniformly scale gradients for all prompts, preserving their relative weights. In contrast, LR/AR fundamentally reshape which prompts receive more gradient signal. For a concrete example from Figure 1 (left panel): at $\rho=⅛$, LR assigns $A^+=7$ while at $\rho=7/8$ it assigns $A^+=1/7$ — a ratio of 49×. GRPO assigns $A^+=2.64$ and $A^+=0.37$ at the same points, a ratio of only 7×. Scaling GRPO's learning rate by any constant preserves this 7:1 ratio. Asymmetric reweighting across difficulty levels is therefore a qualitatively different mechanism from simply taking larger gradient steps.
>
> Thanks again for the careful read. We hope these address your questions.

---

> > ### Author Rebuttal · Reviewer_HZCE · 2026-04-03
> >
> > I thank the authors for the detailed response and additional experiments. Q1 (warmup) and Q4 (LR scaling distinction) are adequately addressed — an actual ablation comparing GRPO with higher learning rates would have been stronger, but I understand the rebuttal timeline constraints. I would appreciate it if the new methods (e.g., Sqrt-R) and warmup ablations are incorporated into the revised paper. I will slightly raise my score.

---

### Official Review · Reviewer_5BWv · 2026-03-10

**Soundness:** 1
**Presentation:** 3
**Significance:** 2
**Originality:** 2
**Overall Recommendation:** 4
**Confidence:** 3

**Summary:**

The paper proposes asymmetric prompt reweighting for Large Language Model Post-Training in reasoning settings, showing that increasing the relative weighting of difficult prompts improves performance in low-success regimes. The proposed re-weighting is theoretically motivated and validated through a set of experiments on simple math tasks on raw pre-trained models. These results are contrasted with experiments on fine-tuned reasoning models, which inhibit higher initial success and thus benefit less from the proposed reweighting.

**Compliance With Llm Reviewing Policy:**

Affirmed.

**Final Justification:**

I appreciate the authors explanation and justification regarding the central claims of the paper. Assuming that there will be some additional explanation in the revision, the approach seems technically sound. The additional experiments further strenghten the claims that are made by the authors.
I'm still hesitant about the scope of the contribution, as the approach targets a narrow domain, but would not be opposed to accepting the paper.

**Key Questions For Authors:**

- Most importantly, how exactly does the presented reweighting enable non-zero gradients for all-zero groups?
- How would the presented approach perform for more common settings, such as, e.g., Qwen3 on DAPO-Math or EURUS-Math/Code. Do the benefits translate between different model sizes?
- To what extend do the experiment specifics, such as, e.g., the warm-up of Line 258 right influence the experimental results?
- The correct response histogram in Figure 3 is interesting. How would this look when compared across methods?

**Limitations:**

The authors very clearly address the scope and limitations of the work. In particular, they highlight that the approach is strongest in low-success regimes, and provide neutral results for post-SFT models, which is highly appreciated.

**Strengths And Weaknesses:**

### Soundness

- -To get this out of the way first: The central claim of the paper is wrong. The authors claim multiple times (e.g., Line 23 right, Line 075 left, Line 181 left) that their re-weighting yields non-zero gradient signals for all-zero groups. While this may be true in a practical implementation, where the average reward $\hat{\rho}$ is offset by some stability constant, this is not what is being presented in the submission. The paper proposes weightings for the base advantage term $r_i-\hat{rho}$ for reward $r_i$, which is trivially zero if all $r_i$ are zero, making the total expression either undefined or zero. This oversight permeates through large parts of the paper. I will assume some equivalent and mathematically correct claim for the remaining review to allow for a neutral evaluation of the provided submission.
- +Except for the shortcoming above, the paper does a good job of theoretically supporting its claims and nicely links its theory to the experimental results
- +The experiments are generally well-designed and cover both raw pre-trained and post-SFT models. The difference between these settings is highlighted well and supports the main claims made in the paper.
- -The experiments are relatively narrow. The authors consider two setups in the low-success regime and another two setups for post-SFT models. However, they only consider a single, relatively uncommon type of model per setup, making the statistical significance of the shown benefits difficult to assess.
- -The GSM8k experiments in Section 4.2 describe a warm-up of the advantages over the first N steps, which is not justified throughout the rest of the paper. While this is a minor detail, this one-off treatment is slightly suspicious and potentially weakens the general contribution of the proposed method.

### Presentation

- +The paper is well-written and clearly structured and motivated.
- +The related work section clearly and accurately distinguishes the contributions of the paper from those of existing work.
- +The paper includes relevant information to reproduce the experimental results.
- +The proposed contribution is clearly targeted towards the low-success regime, making it easy for practitioners to know when to utilize it.
- -The formatting for several of the figures could be improved. For example, Figure 1 has a large whitespace ont he right, and the left subplot of Figure 2 is very narrow and difficult to read. Multiple random seeds for the experiment in Figure 2 are appreciated, but the presentational clarity could be improved by, e.g., presenting the mean and standard deviation rather than three identical-looking plots.
- -The overall Reinforcement Learning notation is slightly off. For example, the paper defines PPO as an off-policy algorithm, which contradicts commonly used definitions in the RL community. While this misclassification seems somewhat common in the RL-LLM literature, and the claims of the paper are largely independent of these details, a consistent nomenclature would still be appreciated and allow for a clearer placement of the presented submission.

### Significance

- +The paper addresses sample weighting in the RL post-training for large language models, a popular and highly relevant field.
- -The practical contribution of the paper lies in proposing a sample reweighting, which is a relatively incremental improvement that likely comprises only a couple of lines of code.
- -The benefit of the proposed changes is limited to low-success regimes, and even there, the experimental results only show modest improvements on selected models, making the impact of the approach difficult to assess.

### Originality

- +The mathematical derivations are interesting and provide some insights into the training dynamics of group-based On-Policy RL algorithms such as GRPO.
- +The proposed re-weighting is well-motivated within this mathematical framework and overall an interesting idea.
- +The paper clearly distinguishes itself from related work.
- -Given the large amount of closely related papers that propose very similar modifications, the novelty of the presented method is relatively limited.
- -The work does not propose new tasks or data, and limited theory and perspectives to advance the field.

---

> ### Author Rebuttal · Authors · 2026-03-31
>
> Thank you for your review and acknowledging that the experiments are well designed and that theory and experiments are well linked. We respond below to each of your questions and points of potential misunderstanding.
>
> **`the main claim of the paper being wrong' [because certain expressions get zero] and Question 1:** Thank you for the careful reading. The concern is understandable when looking at the direction ($d_x$) and the weight ($\omega_x$) in isolation. However, throughout, weights ($\omega_x$) multiply with directions ($d_x$) and what gets multiplied with actual log-gradients of the network are the *effective weights* (i.e., $\omega_x \cdot \rho \cdot (1-\rho)$). **The effective weights are always well defined.** We do *not* offset rho (as suggested). Please see Figure 1 for the effective weights and note that they are in the interval [0,1] for all values of $\rho$, including $\rho=0$. Our central claim is thus not wrong as suggested. We will clarify this potential point of confusion in the paper, thanks for raising this point.
>
> **Note on Warmup and Question 3:** Thank you for your note regarding warmup. Prompted by your question, we have conducted a thorough ablation on GSM8K with warmups of 0, 5, 10, 20, 40, and 100 steps. We found that the 50/100-step warmup in our original submission was overly conservative: even 10 steps suffice, and even training without warmup (0) also converges, just with a slower initial phase. Specifically, no monitoring during training is needed in practice: one can simply evaluate the base model's initial accuracy and apply a brief warmup if it is below around 2-3%.
>
> **experiments are narrow and Question 2:** We have added two additional experiments that cover new reasoning domains. The Hyperphantasia experiment (https://ibb.co/0yHtfL7g) confirms the benefit of asymmetric weighting in a new domain (visual reasoning) with a new model family (vision-language), while K&K confirms our theory's prediction (Proposition 1) that the benefit disappears at moderate initial accuracy (please see response to Reviewer EXiS for details). Together with our math experiments, we now have six settings across three reasoning domains (math, logic, visual reasoning) and three model families, demonstrating that asymmetric weighting provides a general principled method rather than a domain-specific trick.While we did not run the specific Wwen3/DAPO-Math/EURUS settings the reviewer suggests, our post-SFT DAPO-math experiment uses the same dataset, and our new Hyperphantasia experiment demonstrates the method on a vision-language model. Together these cover a broad range of practical settings.
>
> **Significance:**
> * We respectfully disagree that the method is characterized as limited in significance because it can be implemented in a few lines of code changes starting from GRPO. Given the popularity of GRPO and consistent improvements we see, we believe the simplicity of integration into existing RL codebases is a desirable feature adding to the method’s significance.
>
> * We also disagree that the modest improvements make the impact less significant. Improvements of +5% for an RL run are significant, and we would not expect much more through a method improvement in an RL run. For reference, the DAPO paper's modifications over GRPO also comprise a few lines of code and yield comparable improvements, yet are considered significant contributions.
>
> **Question 4:** Figure 3 for GRPO looks comparable for step 5 through 220, but stagnates at step 220. LR continues to improve from Step 220 to 440, i.e., more of the prompts with low success rate improve to higher success rate. This is consistent with LR's stronger gradient signal for these prompts.
>
> We hope these clarifications answer your questions. Specifically, we hope it addresses your primary soundness concern, as **the non-zero gradient claim (the basis for the soundness score of 1) is mathematically correct via the limiting argument above.** We would be grateful if the reviewer reconsidered their assessment in light of this clarification and the additional experiments.

---

> > ### Author Rebuttal · Reviewer_5BWv · 2026-04-02
> >
> > I thank the authors for the comprehensive response. I believe my concerns are sufficiently addressed, and appreciate the authors clarification on the zero-gradient claim, as well as the additional experiments. Regarding the significance, I wouldn't fully agree with the DAPO comparison, since the presented method explicitly targets low-success regimes and thus seems more narrow in its application.
> > I will slightly raise my score to reflect these improvements to the paper.

---

### Official Review · Reviewer_Zg8D · 2026-03-12

**Soundness:** 3
**Presentation:** 3
**Significance:** 3
**Originality:** 3
**Overall Recommendation:** 5
**Confidence:** 3

**Summary:**

This paper studies whether prompts that are harder to solve should be upweighted during training. It shows empirically that emphasizing harder problems is beneficial in from-scratch RL settings, but does not provide the same benefit in post-SFT RL, where the model already achieves a moderate success rate. The paper also supports these findings with theoretical analysis, establishing success-rate dynamics that explain why upweighting harder prompts can reduce the time needed to reach a target success rate.

**Compliance With Llm Reviewing Policy:**

Affirmed.

**Final Justification:**

This paper presents well and is technically sound, all of its claims are supported by theorectical analysis and emprical experiments. Hence, I decided to maintain my positive score.

**Key Questions For Authors:**

1. In post-SFT RL training, there are still some questions on which the model has a very low success rate, and SFT may not help much on those cases. Why, then, would asymmetric prompt weighting still fail to provide a benefit in this setting?

2. Why does asymmetric prompt weighting appear to achieve better final performance than GRPO and show no clear plateau in Figure 2? Could the authors provide more insight into this result? From my reading of Section 5, the analysis mainly explains how weighting affects the dynamics of success rates, but it is less clear how it explains differences in final performance.

3. What assumptions are required to establish the theoretical results in Section 5?

**Limitations:**

Yes.

**Strengths And Weaknesses:**

Strengths: The paper is well presented and easy to follow. Its main claims are supported by both empirical experiments and theoretical analysis, which helps clarify why asymmetric weighting can improve zero-RL-style training. The paper also provides useful insights into the success-rate dynamics of RLVR, particularly in relation to how different prompts are weighted.

Weaknesses: The evaluation is somewhat limited. The from-scratch RL experiments are restricted to relatively simple tasks, including Countdown, multiplication, and GSM8K, and each setting only considers a single model. A more comprehensive evaluation across a wider range of models and tasks would strengthen the empirical support for the paper’s conclusions. As a minor point, the explanation for not using Qwen models in the math training section feels repetitive.

---

> ### Author Rebuttal · Authors · 2026-03-31
>
> Many thanks for the review and positive assessment.
>
> Regarding the somewhat limited evaluation setup: To further evaluate the benefits of asymmetric prompt weighting and in particular how they extend beyond mathematical reasoning, we conducted a from-scratch RL experiment for vision as well as an additional experiment on a logical deduction task; please see the response to reviewer EXiS. Together with the original experiments, we now cover six settings across three reasoning domains (math, logic, visual reasoning), three model families (Qwen, Llama, Qwen-VL), and both from-scratch and post-SFT regimes.
>
> Below, we answer to your questions:
>
> **1. Remaining low success rate questions:** We think that those remaining questions lie beyond the model's capabilities that are to a large extent determined by the pre-training. Unlike in from-scratch RL, where the model traverses a wide accuracy range and most prompts are eventually solvable, in post-SFT RL the hard prompts remain unsolvable also for asymmetric prompt weightings.
>
> **2. Plateau and insights on Fig. 2:** In the meantime, we did continue training beyond the 500 steps shown in Figure 2, and the model does plateau there. We agree that the theory does not predict the difference in plateau levels between the asymmetric weightings and symmetric ones: under the abstractions (optimization over the policy and population-level objectives) of Section 5.1, the optimal policy for all schemes reaches $\rho = 1$. These standard assumptions suffice to demonstrate the effect of different weights and optimality of LR in the low-success regime, which is our primary theoretical focus. Relaxing them (e.g., incorporating parameterization over model parameters) is a promising direction for explaining the observed plateau differences, and we will add this to the limitations and future work section.Thank you for the note.
>
> **3. Theory assumptions:** The theoretical results in Section 5 rely on three assumptions: (i) optimization over policies rather than parameters, (ii) a population-level objective ($M\gg 1$ rollouts), and (iii) continuous-time dynamics. As mentioned in Sec. 5.1, Assumptions (i) and (ii) are standard abstractions in the literature. Assumption (iii) of Sec. 5.2 allows us to cast population-level policy optimization as a concrete ODE, which can be solved in closed form for a given ω or optimized over the choice of $\omega$, as done in our theorems.

---

> > ### Author Rebuttal · Reviewer_Zg8D · 2026-04-03
> >
> > Thanks authors for the comprehensive response and the extra experiements to make the paper more completed. I decide to maintain my score.

---

### Official Review · Reviewer_EXiS · 2026-03-12

**Soundness:** 2
**Presentation:** 3
**Significance:** 3
**Originality:** 2
**Overall Recommendation:** 4
**Confidence:** 4

**Summary:**

This paper studies prompt-level weighting in LLMs RLVR, arguing that common RLVR methods such as GRPO and RLOO effectively emphasize prompts with intermediate success probability while downweighting very hard and very easy prompts.  Building on this view, the authors propose asymmetric prompt weightings that assign larger weights to low-success prompts with the goal of making training focus more on currently failing questions.  A notable feature of these weightings is that they can produce nonzero learning signals even when all samples for a prompt are incorrect, unlike standard GRPO-style updates. Empirically, the paper evaluates these methods on two from-scratch RL settings and two post-SFT settings, and finds that asymmetric weighting is most helpful in low-success, from-scratch regimes, while providing little advantage once the model already starts from a relatively strong instruction-tuned baseline.  On the theory side, the authors derive success-rate dynamics for prompt-weighted RLVR and show that the preferred weighting depends on how training time is measured.  In particular, they argue that under a rollout-cost-adjusted notion of effective time, low-success prompts should be upweighted, and that the proposed log-style weighting is asymptotically optimal in the low-success regime.  Overall, the paper’s main contribution is to identify asymmetric prompt weighting as a useful design principle specifically for low-success RLVR training, and to support this claim with both experiments and a simple theoretical framework.

**Compliance With Llm Reviewing Policy:**

Affirmed.

**Final Justification:**

My final recommendation is weak accept, though still somewhat borderline. The paper is clear, technically solid, and appropriately modest in its claims. I found the main empirical conclusion reasonably well supported: asymmetric prompt weighting is helpful in low-success, from-scratch RLVR settings, but provides much less benefit in post-SFT regimes. I also found the regular-time versus effective-time analysis to be the most interesting part of the paper, and the most original contribution.

My main reservations remain about originality and significance. The overall framing feels somewhat incremental relative to prior objective/rescaling views of RLVR, and the practical impact seems specialized rather than broad. So I view the paper more as a targeted optimization insight than as a broadly new algorithmic contribution.

My biggest concern was the stability of nonzero updates for zero-success prompts. The rebuttal addressed this concern meaningfully through additional ablations and experiments, which increased my confidence in the paper’s soundness and made me more positive overall. As a result, I raised my score.

**Key Questions For Authors:**

1. My biggest concern is, see "soundness", could the authors provide a more systematic discussion of the stability of nonzero updates for zero-success prompts?

2. The paper argues for both effects: one is “do not ignore very hard prompts,” and the other is the more specific choice to keep updating even when \hat\rho=0. Can the authors better characterize how much of the observed gain comes specifically from allowing zero-success prompts to contribute nonzero gradients, versus more generally upweighting low-success prompts?

**Limitations:**

yes

**Strengths And Weaknesses:**

On soundness, I think the main empirical claim is supported reasonably well: the authors are careful to distinguish between from-scratch, low-success regimes, where asymmetric weighting appears helpful, and post-SFT regimes, where the benefit is much smaller or absent. That claim is consistent with both the experiments and the way the paper frames its contribution, and I appreciated that the authors did not overclaim universality. The theory of success-rate dynamics is also a strength: this analysis is clean, easy to follow, and the distinction between regular optimization time and rollout-cost-adjusted effective time is insightful.
At the same time, there are some soundness-related weaknesses. The most important one, in my view, is the treatment of zero-success prompts with nonzero updates. This is one of the paper’s most distinctive ideas, and the authors explicitly emphasize that their method can still produce a learning signal when all sampled completions are incorrect. The experiments also suggest this mechanism can help. However, the stability implications of this choice are not analyzed in much depth. The authors mention it in chapter4.2:
"In this regime, most prompts have no correct completions (ρ = 0), and dominate the gradient for AR and LR. We therefore set the advantages of AR and LR for prompts with success rate ρ = 0 to 0 for a warmup period of 50/100 steps for batchsizes 256/128."
But the paper does not thoroughly characterize when it is stable, when warmup or clipping is necessary, or how sensitive performance is to the prevalence of zero-success prompts. That omission matters because this is precisely the regime where the method is supposed to be most useful.

On presentation, I think the paper is clearly written and easy to follow. The overall narrative is coherent: first unify existing binary-reward RLVR methods through prompt weighting, then propose asymmetric alternatives, then test them in two different training regimes, and finally explain the observed behavior through a simple theory.

On significance, I see this as a useful but somewhat specialized contribution. The problem of zero-success prompts is relevant in RLVR. But the scope of this paper seems narrow. The gains appear regime-dependent rather than broad, and the paper itself acknowledges that the main benefit is concentrated in from-scratch training. So while I do think the paper advances understanding, I view it more as a targeted optimization insight than as a broadly algorithmic contribution.

On originality, I would again say the paper is mixed. I do not find the overall framing highly original, since much of the “different RLVR methods correspond to different rescalings/objectives” perspective is already present in prior work, including the cited “What is the objective of reasoning with reinforcement learning?” paper. The proposed methods themselves feel like relatively simple modifications within that lens: instead of symmetric weighting that emphasizes ambiguous prompts, choose asymmetric weighting that favors low-success prompts. That said, originality does not have to mean inventing an entirely new algorithmic family, and I do think the paper contributes a fresh insight: the regular-time versus effective-time distinction gives a clean theoretical rationale for why one might prefer GRPO-like weighting in one regime and LR-like weighting in another. That part deepens understanding rather than merely proposing a heuristic, and it was the most original aspect of the paper for me.

---

> ### Author Rebuttal · Authors · 2026-03-31
>
> Many thanks for your review and for noting the clarity, strong support for the claims, and originality of the theory.
>
> Regarding the treatment of zero-success prompts, in particular the stability implications of this choice, and necessity of warmup:
>
> We conducted ablations on GSM8K with warmups of 0, 5, 10, 20, 40, and 100 steps. We find that the zero-success signal does not cause instability: even without warmup, training succeeds and achieves end-performance that is at least as good as that of GRPO despite having an initial slow-phase over the first roughly 80 steps before it starts climbing rapidly after that. Furthermore, a brief warmup of just 10 steps eliminates entirely this slow phase and leads to the performance gains we report in the original submission with the 100-step warmup, which  was indeed overly conservative. Thus, in practice, we recommend a short warmup of 10–20 steps when the base model's initial success rate is very low (<1.5%), which is easy to assess before training. Intuitively, when >=99% of prompts have ρ=0, the zero-success gradient, which pushes away from incorrect completions without positive directional signal, dominates the update, providing less informative gradients. But as more prompts achieve ρ>0, e.g., after a small warm-up this zero-success gradient signal becomes a useful complement.
>
> We also added two more experiments, described below, both covering new reasoning domains.
>
> The Hyperphantasia experiment confirms the benefit of asymmetric weighting in a new domain (visual reasoning) with a new model family (vision-language), while K&K confirms our theory's prediction that the benefit disappears at moderate initial accuracy. We now have six settings across three reasoning domains (math, logic, visual reasoning) and three model families, demonstrating that asymmetric weighting provides a general principled method rather than a domain-specific trick.
>
> Experiment 1: We carry out further experiments on the Hyperphantasia [1] ball-trajectory task at two difficulty levels: medium and hard difficulty. The figure https://ibb.co/0yHtfL7g compares GRPO and LR when training Qwen2.5-VL-7B on the Hyperphantasia ball-trajectory task at two difficulty levels. On medium-difficulty problems, LR improves much more rapidly, reaching high validation accuracy earlier and maintaining a more stable trajectory, while GRPO progresses more slowly and exhibits larger fluctuations before eventually catching up; both methods ultimately attain essentially perfect peak performance, with a maximum accuracy of 100.0. On hard-difficulty problems, the same qualitative pattern persists: LR again learns faster and more consistently, and it also achieves a higher final peak accuracy, reaching 97.7 compared to 94.7 for GRPO. Overall, these results suggest that LR provides a clear optimization advantage for visual reasoning.
>
> [1] Shahab et al., Hyperphantasia: A Benchmark for Evaluating the Mental Visualization Capabilities of Multimodal LLMs, Neurips 2025.
>
> Experiment 2: To further evaluate whether the benefits of asymmetric prompt weighting extend beyond mathematical reasoning, we conduct an additional from-scratch RL experiment on a logical deduction task. We train Qwen2.5-7B on Knights and Knaves (K&K) logic puzzles from the K&K dataset, a constraint satisfaction task requiring propositional reasoning rather than arithmetic computation. The dataset consists of 2,700 training examples, and the model receives a binary reward of 1 only if all character identities are correctly identified, and 0 otherwise. GRPO and the considered asymmetric advantage estimators perform comparably, consistent with our theoretical prediction the initial performance of ~0.15 places this setting in the moderate-success regime where GRPO's symmetric weighting is near-optimal. This provides further evidence that the benefit is regime-dependent.
>
> Regarding characterizing whether the gains come from, zero-success gradients vs general upweighting: Both assigning non-zero weight to prompts with ρ = 0 and upweighting low-success prompts is important for performance. To demonstrate this, we performed further TinyZero experiments for a variant of AR, for which we assign zero weight to zero-success prompts (i.e.,groups with ρ = 0). We find that with the same stepsize, training is less stable, and leads to worse performance (about 3% worse than original AR, but still 3% better than GRPO). A smaller stepsize fixes the instability, but results in worse performance. As a further experiment, we considered the weight ω = p(1 − ρ)/ρ, which is another example of an asymmetric weight that assigns zero weight to zero-success groups. We call this kimi-weighting since the prioritized sampling strategy used in the paper ``Kimi K1.5: Scaling Reinforcement Learning with LLMs’', where samples are reweighted by 1−ρ, which corresponds to that weight. Kimi-weighting performs worse than the other asymmetric weightings.

---

> > ### Author Rebuttal · Reviewer_EXiS · 2026-04-03
> >
> > After reading the rebuttal, I am more positive about the paper and will raise my score. The additional ablations on warmup length meaningfully address my main concern about the stability of nonzero updates for zero-success prompts, and the new experiments strengthen the paper’s regime-dependent empirical story across more domains. My concerns about limited novelty and somewhat specialized significance still remain, but overall the rebuttal increases my confidence in the paper’s technical soundness and practical relevance.

---

> > > ### Author Response · Authors · 2026-04-05
> > >
> > > We’re glad the rebuttal helped address your main concerns, especially on the stability issue and the broader empirical support. We also appreciate your increased confidence in the paper’s technical soundness and practical relevance.
> > >
> > > We certainly agree there are still more exciting opportunities to explore on the algorithmic front of RLVR optimization. At the same time, we believe the paper makes a meaningful step beyond a specialized empirical finding by providing a broader framework for understanding when different weighting rules are preferable, supported by both theory and experiments across multiple regimes.
> > >
> > > We appreciate your thoughtful feedback, and would be grateful if your score could be updated accordingly, as you mentioned.

---

### Decision · Program_Chairs · 2026-04-30

**Decision:**

Accept (regular)

**Comment:**

The paper studies asymmetric prompt weighting for RL with verifiable rewards and argues that upweighting low-success prompts is especially useful in low-success, from-scratch RL regimes. Reviewers agreed that the paper is clearly written, technically well presented, and supported by a useful theoretical perspective that helps explain when asymmetric weighting should be preferred over GRPO-style symmetric weighting.

The main discussion centered on the treatment of zero-success prompts and whether the resulting learning signal is sound and stable. As mentioned by reviewers EXiS, 5BWv, and HZCE, this was the key concern in the initial reviews. After reading the rebuttal and added experiments, these concerns were substantially addressed. In particular, the additional warmup ablations and broader empirical evidence clarified the behavior of the zero-success signal and increased confidence in the paper’s technical soundness.

Reviewers also noted limitations. The empirical gains are concentrated in from-scratch low-success regimes, with little benefit in post-SFT settings, and the overall contribution is somewhat targeted rather than broad. The evaluation breadth could also be stronger. That said, the paper is careful about this scope, does not overclaim, and provides a coherent combination of analysis and experiments supporting a useful insight for RLVR training.

Overall, I recommend acceptance. The paper makes a technically solid and well motivated contribution, and the rebuttal successfully resolved the main soundness concerns while strengthening the empirical case.